# Crystallization of carboplatin-loaded onto microporous calcium phosphate using high-vacuum method: Characterization and release study

**Cristiane Savicki** [ORCID] ☯ *, **Nelson Heriberto Almeida Camargo** ☯ †, **Enori Gemelli** ☯

Department of Mechanical Engineering, College of Technological Science, Santa Catarina State University, Joinville, Santa Catarina, Brazil

☯ These authors contributed equally to this work.
† Deceased.
* cristianesavicki@gmail.com

**Data Availability Statement:** All relevant data are within the manuscript and its Supporting Information files.

## Abstract

Drug delivery systems are a new approach to increase therapeutic efficacy and to reduce the side effects of traditional treatments. Calcium phosphates (CaPs) have been studied as drug delivery systems, especially in bone diseases. However, each system has some particularities that depend on the physical and chemical characteristics of the biomaterials and drug interaction. In this work, granulated CaPs were used as a matrix for loading the anticancer drug carboplatin using the high-vacuum method. Five compositions were applied: hydroxyapatite (HA), β-tricalcium phosphate (β-TCP), biphasic HAp 60%/β-TCP 40% (BCP), β-TCP/MgO nanocomposite, and β-TCP/SiO$_2$ nanocomposite. Carboplatin drug in 50, 60, and 70 mg/g was precipitated on the surface of CaPs. Morphological, chemical and surface modifications in the carboplatin-CaPs were investigated by scanning electron microscopy (SEM), energy dispersive spectroscopy (EDS), backscattered electron microscopy (BSE), X-ray diffraction (XRD), X-ray fluorescence spectroscopy (XRF), Fourier transform infrared (FT-IR), and Raman spectroscopy. The characterization of the CaP-carboplatin biomaterials showed heterogeneous crystalline precipitation of the drug, and no morphological modifications of the CaPs biomaterials. The *in vitro* release profile of carboplatin from CaPs was evaluated by the ultraviolet-visible (UV-Vis) method. The curves showed a burst release of upon 60% of carboplatin loaded followed by a slow-release of the drug for the time of the study. The results were typical of a low-interaction system and physisorption mechanism. The high-vacuum method permitted to load the high amount of carboplatin drug on the surface of the biomaterials despite the low interaction between carboplatin and CaPs.

## Introduction

Calcium phosphate (CaP) biomaterials are used as bone substitutes and as materials in the regeneration tissue process due to their similarities with natural apatites. These bioceramics

**Funding:** This study was financed in part by the Coordination for the Improvement of Higher Education Personnel – Brasil (CAPES) – Finance Code 001 - https://capes.gov.br, and Foundation for the Support of the Scientific and Technological Research of Santa Catarina State (FAPESC) - www.fapesc.sc.gov.br. The funders had no role in study design, data collection and analysis, decision to publish, or preparation of the manuscript.

**Competing interests:** The authors have declared that no competing interests exist.

present strong physicochemical similarity with the mineral part of bone tissue, resulting in favorable properties for their use as a filler in bone and dental defects [1,2]. CaP bioceramics are recognized as biocompatible biomaterials; they can be osteoconductive and osteoinductive. The biological behavior of the biomaterial is deeply connected to chemical, microstructural, and surface properties, as well as to porosity [2,3]. Furthermore, these characteristics influence directly the physisorption/chemisorption processes that permit loading the CaP microstructure with therapeutic substances. As a result, suitable characteristics of CaPs permit its use as a matrix for drug-delivery system [4]. The amount of drug in CaP and the release behavior are influenced by the size, shape, surface area, and crystallinity of the biomaterials. These characteristics can be achieved or modified by the material source, chemical composition, synthesis method, and processing conditions used in biomaterial acquisition [5–7]. CaPs can be used in different forms, depending on the desired application [8,9]. CaP powders, cements, blocks, scaffolds or granules, as well as CaP composites with natural and synthetic polymers, can be used as a matrix for drug loading [10–12]. Biodegradable CaPs nanopowders or composites have been studied as alternatives nanocarriers or prodrugs to traditional systemic therapies [13–17]. However, the main application of CaPs as drug delivery systems is to release drugs in bone diseases. CaPs have been widely used as bone fillers in orthopedic and maxillofacial applications, osteosarcoma, and osteoporosis. The addition of drugs in CaPs biomaterials to be delivered locally can prevent or treat bone diseases and accelerate tissue recovery [18–21].

Hydroxyapatite, tricalcium phosphate, biphasic materials, and substituted apatites have been used as CaP biomaterials in bone treatment and delivery systems [7,22]. CaPs also can form systems used to coat metallic implants that increase bioactivity and drug delivery [23].

Several substances have been loading in CaP delivery systems: antibiotics and anti-inflammatories, to prevent and treat osteomyelitis [24–27]; drugs and hormones, to improve the bone remodeling process [7,28]; genetic material [29,30]; and therapeutic drugs, in cancer disease [22,31,32]. Cancer treatment usually involves undesirable side effects. Local delivery of therapeutics in oncology is targeting benefits, such as reducing the doses of the drug, minimizing and preventing side effects and cytotoxicity in other organs and tissues [31–33].

In the last decades, many systems were designed using CaPs, namely hydroxyapatite, to deliver antitumoural drugs [34–38]. Platinum-based agents, such as cisplatin, carboplatin, kiteplatin, and others have received attention in systems with CaPs [14,32,36,39], to provide therapeutic agents specifically into cells and tissues, including in bone tumors, by combining the bone remodeling characteristics of CaPs with their suitable features to be used as delivery systems. Designing a drug delivery system should consider the interactions between the drug and carrier matrix. Different types of interaction can rule drug loading and release. Depending on the affinity of the drug with CaP surface and its characteristics, the predominant interaction mechanism verified may be electrostatic or hydrophobic/hydrophilic physisorption or chemical interactions, which modify the drug loading amount and the release time of the drug [4,22]. The amount of the substance loaded in biomaterials for drug delivery depends on the material used, the method to drug loading, and the final application. The parameters of the CaP used as a carrier and the drug, as well as the drug loading process, result in different systems that present specific characteristics of delivery behavior.

In the literature, many drugloading techniques were reported. The solvent evaporation was used to remove toxic substances used in the case of poorly water-soluble drugs or polymer-drugs systems [40,41]. Drugs or molecules have been loaded in porous biomaterials by impregnation methods, based on adsorption processes that depend on the interaction between drug and matrix after a time of contact. It has been reported the use of pressure to improve the loaded amount of drugs in porous matrices [42–44]. The study of Itokazu et al. [35] used the vacuum pressures to improve the penetration of an anticancer drug in calcium

phosphate blocks. In this paper, the vacuum method was applied to fill the interconnected microporosity with the drug solution and promote the deposition of carboplatin through solvent volatilization. The carboplatin drug was loaded in well-crystalline granular CaP biomaterials by high-vacuum pressure. Hydroxyapatite and (HA) and β-tricalcium phosphate (β-TCP) powders were synthesized by the wet chemical method using calcium carbonate ($CaCO_3$) and phosphoric acid ($H_3PO_4$). Granulated biomaterials were obtained by attrition milling. Five different compositions were tested: hydroxyapatite (HA), β-tricalcium phosphate (β-TCP), biphasic HA 60%/β-TCP 40% (BCP), β-TCP/MgO nanocomposite, and β-TCP/SiO$_2$ nanocomposite. HA, β-TCP and BCP bioceramics are widely studied as biomaterials for bone regeneration and delivery systems. The β-TCP nanocomposites were developed to achieve a higher porosity than the β-TCP matrix. The drug loading and release were carried out to evaluate the influence of microstructure and chemical composition on the CaPs-carboplatin systems. Characterization studies and release test in the CaPs-carboplatin biomaterials were carried out to determine the interaction between the matrices and the drug and the *in-vitro* release profile of the system.

## Materials and methods

### Materials

Calcium carbonate ($CaCO_3$) from Synth, batch number 63767, and phosphoric acid $H_3PO_4$−85 wt% in water) from Dinamica, batch number 68436, were used in the synthesis of granular biomaterials. Carboplatin was obtained from pharmaceutical forms Tevacarbo solution for injection, batch number 12B14LA, and Tecnocarb powder for solution, batch number 85922. Amorphous silica powder and magnesium oxide from magnesium carbonate calcination were used to produce granular nanocomposites.

### Synthesis and preparation of granular biomaterials

HA and β-TCP were prepared by the wet chemical method, modified from a previously published study [45], using calcium carbonate ($CaCO_3$) and phosphoric acid ($H_3PO_4$) in the quantities required for the formation of precipitates with the desired Ca/P molar ratio. Calcium carbonate was dispersed in distilled water and sonicated in an Ultrasonic Processor 500 W; 20 kHz; 13 mm probe; Sonics Vibra-Cell VCX 500 for 10 minutes. Phosphoric acid was slowly added dropwise into $CaCO_3$ suspension under stirring. The pH of the reaction medium was measured throughout the synthesis. The suspension was kept under stirring overnight and dried in a rotatory evaporator. In this work, it was not used any buffer or alkaline solution, such as ammonia hydroxide ($NH_4OH$), as a pH adjustment agent, as reported in other works [46–48]. Powders of hydrated calcium phosphate precipitated in the synthesis were sieved on a 100 μm mesh. They were calcined at 900˚C for 2 hours, and the obtained nanostructured powder of HA and β-TCP were used to produce granulate biomaterials. Five different compositions of granular biomaterials were produced by an attrition milling, in a method described in other studies [2,49]: HA, β-TCP, a biphasic biomaterial with ratio 60/40 in volume and two nanocomposites, β-TCP/SiO$_2$ 5% and β-TCP/MgO 1%, in volume. Briefly, calcined powders and the second phase for nanocomposites were added in a Netzsch attrition mill with a 50/50wt% solid/liquid concentration in ethyl alcohol and 2.5 mm zirconium spheres, at 540 rpm for 1 hour. After the process, the material was dried in a rotatory evaporator, and the granular biomaterials recovered were sieved and sintered at 1,100˚C for 2 hours.

## Carboplatin loading on granular biomaterials

Carboplatin solution in a drug concentration of 10 mg/g was obtained from pharmaceuticals forms that contain the drug and the same quantity of mannitol in their formula. The solution was added to the biomaterials in the quantities required to achieve three different final concentrations of carboplatin: 50, 60, and 70 mg/g. One g of the granular biomaterial was immersed in the carboplatin solution under $10^{-2}$ millibar pressure vacuum for about 48 hours, until completing the dryness and reaching a constant weight. The weight of the biomaterials after the drug load was verified and compared with the theoretical ones to give the percentage of remaining CaP-carboplatin loaded after the process, by the Eq 1:

$$Remaining\ weight\ (\%) = 100\ x\ W2/W1 \tag{1}$$

In which:

W1 = theoretical weight calculated by the actual weight of the biomaterials and the drug content added, considering carboplatin and mannitol in its formulation;

W2 = weight of the CaP-carboplatin samples after the drug loading.

## Carboplatin release from the granular biomaterials

*In-vitro* release of carboplatin was carried out in phosphate buffer pH 7.4 as dissolution medium, and 250 mg of CaP-carboplatin granular biomaterial was dispersed in 100 mL of phosphate buffer and kept under 80–100 rpm rotation and 37 ± 1˚C. The pH of the medium at 37˚C was measure before and after the release study. The volume of the dissolution medium was calculated in order to guarantee sink conditions in the study, avoiding supersaturation and concentration gradient in the system. Five-mL samples were withdrawn by a pipette and filtered at the times points of 15, 30, 45, 60, 90, 120, 150, 180, 210, 240, 270, and 300 minutes. The sample volume was replaced each time with the same quantity of fresh phosphate buffer to maintain the volume constant. The drug concentration and the carboplatin calibration curve (S1 Fig) were determined by ultraviolet-visible (UV-Vis) absorption spectroscopy, in a Spectroquant Pharo 300M spectrometer, at 232 nm. Absorption values were corrected by CaP biomaterials blank experiments, by subtracting the absorbance of carboplatin free absorbance at the same wavelength. The final concentration in which point was calculated correcting the carboplatin concentration in the previous step in the release test, by the Eq 2 [25]:

$$Ctcor = Ct + \frac{v}{V}\sum_{0}^{t-1} Ct \tag{2}$$

In which:

Ctcor = the corrected concentration of carboplatin in the time *t*;

Ct = the apparent concentration at the time *t*;

v = the volume of the sample in each time point;

V = the total volume of the test medium.

## Biomaterials characterization

Biomaterials characterizations were carried out using different techniques. The microstructure of the biomaterials and the surface morphological features after drug loading were determined by field emission scanning electron microscopy (FE-SEM) technique with a JEOL JSM-6701F

equipment, using a secondary electrons image and backscattered electron micrography (BSE), with the electron acceleration voltage of 15 kV. FE-SEM coupled with an X-ray analysis system (EDS) was used to analyze the elemental composition in the CaP-carboplatin granulated biomaterials. All samples were sputtered with gold-palladium under argon atmosphere.

The specific surface area was measured through Brunauer-Emmett-Teller (BET) method using a Micromeritics ASAP2020 after degassing the granulated biomaterials by $N_2$ gas at 300˚C for 2 hours.

Phase composition was determined by X-ray diffraction (XRD), before and after the drug loading. The studies were performed in a Shimadzu X-ray diffractometer XRD-6000 with CuK$\alpha$ radiation ($\lambda$ = 1.54060 Å), generated at 40 kV/3 mA current intensity, using the diffraction angle of 2$\theta$; step size 0.02˚; between 10–70˚.

The chemical composition and the amount of platinum element in the CaP-carboplatin granulated biomaterials were determined by X-ray fluorescence spectroscopy (XRF) in a Shimadzu EX-720 X-ray spectrometer using rhodium anode X-ray tube.

Vibrational modes of carboplatin were determined in the samples in a Fourier transformer Shimadzu Prestige-21 spectrometer. The Fourier transform infrared (FT-IR) spectra were recorded from 4,000 to 400 cm$^{-1}$ and KBr pellets. Raman spectra were obtained using a Brüker Senterra spectrometer equipped with a He-Ne laser operating at 532 nm and 1,200 L.mm$^{-1}$ grating.

## Results

### Synthesis of calcium phosphate biomaterials

The pH of the suspension was measured during the aqueous precipitation step of the synthesis. The initial pH of the calcium carbonate suspension was about 10 before the addition of $H_3PO_4$ and slightly increased with ultrasonic irradiation application. After the addition of $H_3PO_4$, the pH decreased to about 6 due to the acidification of the reaction medium since no one alkaline solution was added. After the end of the acid addition, the pH increased progressively, reaching the values of 7.1 for CaP 1.67 molar and 7.0 for CaP 1.5 molar in 24 hours (S2 Fig). Nanometric powders were recovered after drying in a rotatory evaporator. After calcination at 900˚C for 2 hours, hydroxyapatite and β-TCP crystalline powders from CaP 1.67 and CaP 1.5 synthesis, respectively, were obtained (XRD and FT-IR characterization in S3 and S4 Figs). The powders presented interconnected microporosity (SEM characterization in S5A and S5B Fig). They were used to produce granulated microporous biomaterials by attrition milling technique.

### Microstructure of the biomaterials

The surface microstructure of granulated biomaterials before drug loading performed by SEM showed that each biomaterial had regular morphology with the presence of interconnected porosity (Fig 1). The HA biomaterial (Fig 1A) had a very-fined microstructure compared with β-TCP (Fig 1B), which presented large-size grains. The biphasic biomaterial (Fig 1C) had the same microstructure with an intermediate size, due to the presence of HA and β-TCP in its composition. The secondary phases added in β-TCP in nanocomposites biomaterials changed the microstructure. β-TCP/MgO (Fig 1D) displayed very-fined globular grains compared with β-TCP only. β-TCP/SiO$_2$ (Fig 1E) presented nanosized structures in the interconnected microstructure, due to the SiO$_2$ added as a secondary phase.

The surface analysis was performed by BSE-SEM to evidence carboplatin in the biomaterials, due to the proportional intensity of backscattered electron emission with the increase in the atomic number. Carboplatin drug was precipitated in different sizes on the surface of the

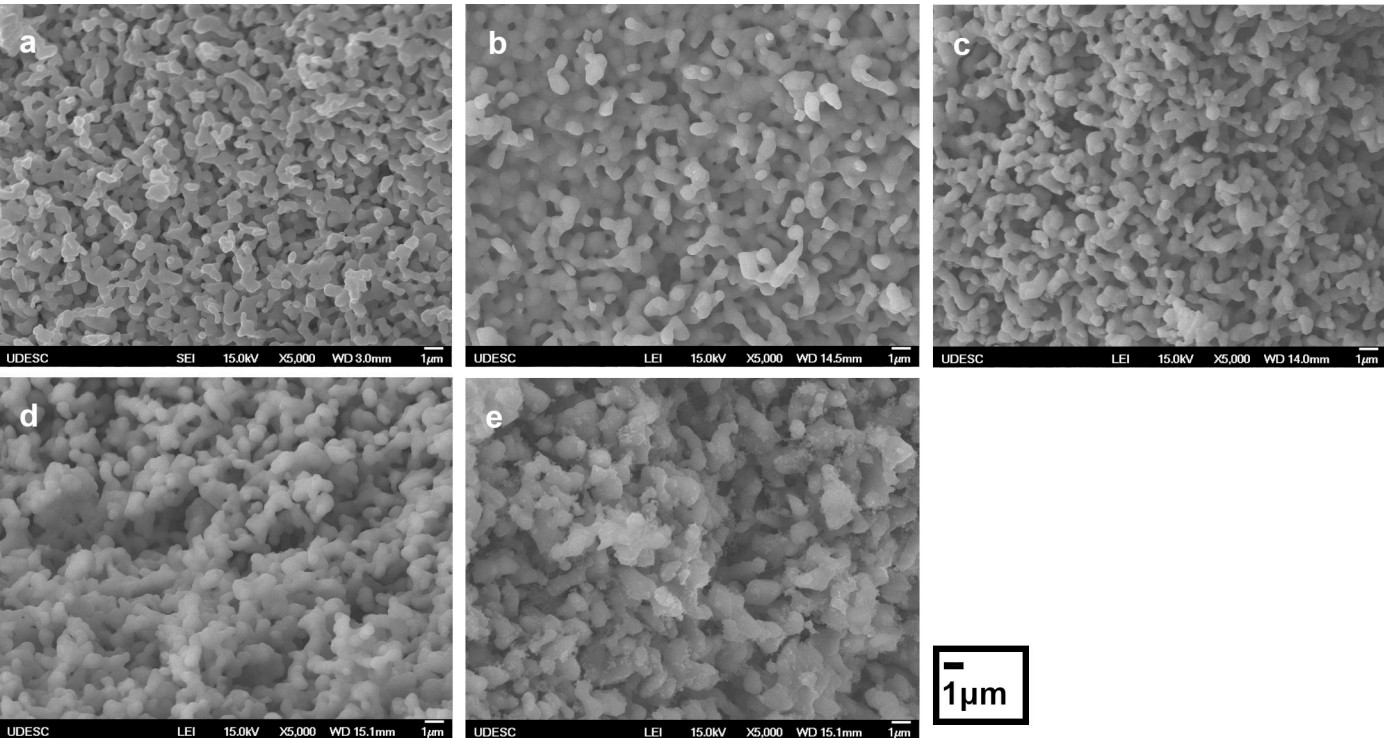

**Fig 1. The initial microstructure of the biomaterials.** Scanning electron microscopy analysis of (a) hydroxyapatite (HA); (b); β-tricalcium phosphate (β-TCP); (c) biphasic calcium phosphate (BCP); (d) β-TCP/MgO nanocomposite; (e) β-TCP/SiO$_2$ nanocomposite.

microporous biomaterials (large arrows). In the BSE image in Fig 2A and 2C, the surfaces of HA and BCP presented micrometric precipitates of carboplatin. Fig 2B shows drug precipitates about 1 μm. Fig 2D shows a structure with different contrast in β-TCP/MgO nanocomposite (straight arrow) that is compatible with mannitol in carboplatin drug, whose carbon structure presents a different behavior in backscattered electron analysis. Fig 2E shows an amorphous precipitation in β-TCP/SiO$_2$ nanocomposite (large arrow). The microstructure of biomaterials after the drug loading process did not change, compared with the previous SEM analysis.

SEM-EDS analysis was carried out to detect the elemental composition of the biomaterials after drug-loading. Fig 3A shows a micrometric precipitate of carboplatin in HA biomaterial. EDS analysis indicated the presence of the platinum element, as well as calcium and phosphorus from the CaP matrix (Fig 3D). Fig 3B shows embedded carboplatin in biphasic CaP matrix. EDS analysis of point 1 (Fig 3F) detected platinum element, while in point 2 only calcium and phosphorus were detected (Fig 3E). Fig 3C shows a drug precipitate without well-defined boundaries, as seen in other materials, and silica nanoparticles were detected in carboplatin precipitates in TCP-β/SiO$_2$ nanocomposite. Elemental analysis of a precipitate in TCP-β/SiO$_2$ nanocomposite (Fig 3G) detected calcium, platinum, and silicon. Palladium element in Fig 3E and 3F came from gold-palladium layer deposited by ion sputtering on the samples before analysis.

## Surface area

The results of BET analysis are shown in Table 1. HA and β-TCP/SiO$_2$ biomaterials presented a greater superficial area, both having a surface area about 2.5 times than the one of β-TCP biomaterial. BCP biomaterial had an intermediate surface area compared with HA and β-TCP.

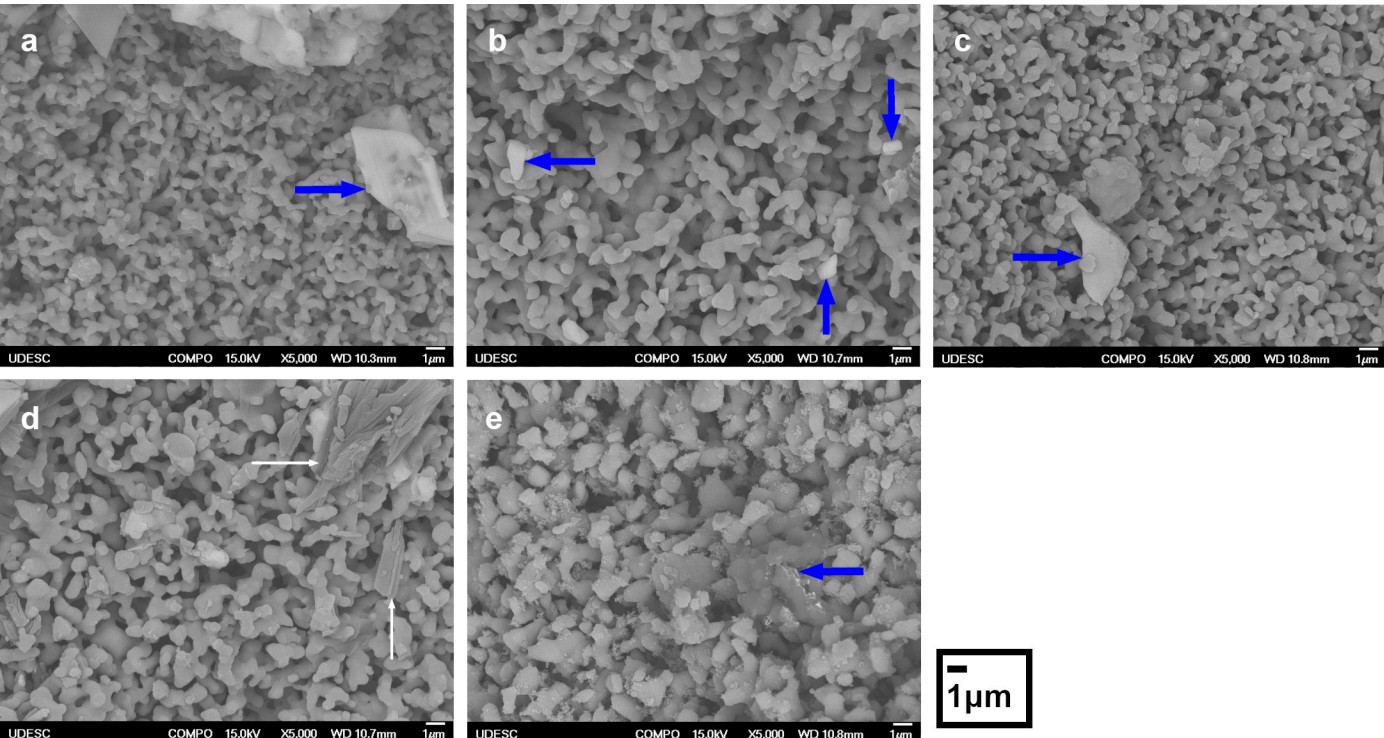

**Fig 2. Morphology of precipitated drug onto calcium phosphate surface.** Backscattered electron microscopy analysis of the biomaterials-carboplatin loading in the 70 mg/g concentration showing the micrometric carboplatin precipitate in (a) hydroxyapatite (HA), (b) β-tricalcium phosphate (β-TCP) and (c) biphasic calcium phosphate (BCP); mannitol precipitate in (d) β-TCP/MgO nanocomposite; amorphous precipitate in (e) β-TCP/SiO₂ nanocomposite.

The addition of a secondary phase in the β-TCP/MgO nanocomposite increased the surface area compared with the β-TCP matrix alone.

## X-ray diffraction (XRD)

The XRD analysis in CaP granular biomaterials before drug loading revealed the high crystallinity of sintered materials (Fig 4A). The high-intensity peaks of HA—JCPDS card ($Ca_{10}(PO_4)_6OH_2$), ref. no. 75–0565—and β-TCP—JCPDS card ($Ca_3(PO_4)_2$), ref. no. 09–0169—are shown in the diffractograms. The XRD patterns from the biomaterials after loading 70 mg/g carboplatin did not indicate any changes in the diffraction pattern of the biomaterials, except for the presence of peaks of carboplatin in all the samples (Fig 4B). The nanocomposite β-TCP/SiO₂ presented carboplatin peaks with low intensity compared with other materials. The drug pattern showed the presence of carboplatin and mannitol in the material. Mannitol presented weak intensity peaks in the drug and was not detected in the biomaterials after drug loading.

## X-ray fluorescence spectroscopy (XRF)

XRF of CaP-carboplatin biomaterials was used to determine the elemental chemical composition and measure the amount of platinum element after drug loading (Table 2). The elements Ca and P were expected in CaP biomaterials. Silicium element was identified in β-TCP/SiO₂ nanocomposite, while Zr and Fe detected in minor quantities were from the zirconium balls and chemical composition of the $CaCO_3$ used in the synthesis, respectively. The platinum element was identified in all composition and doses. The quantities measured did not exhibit a

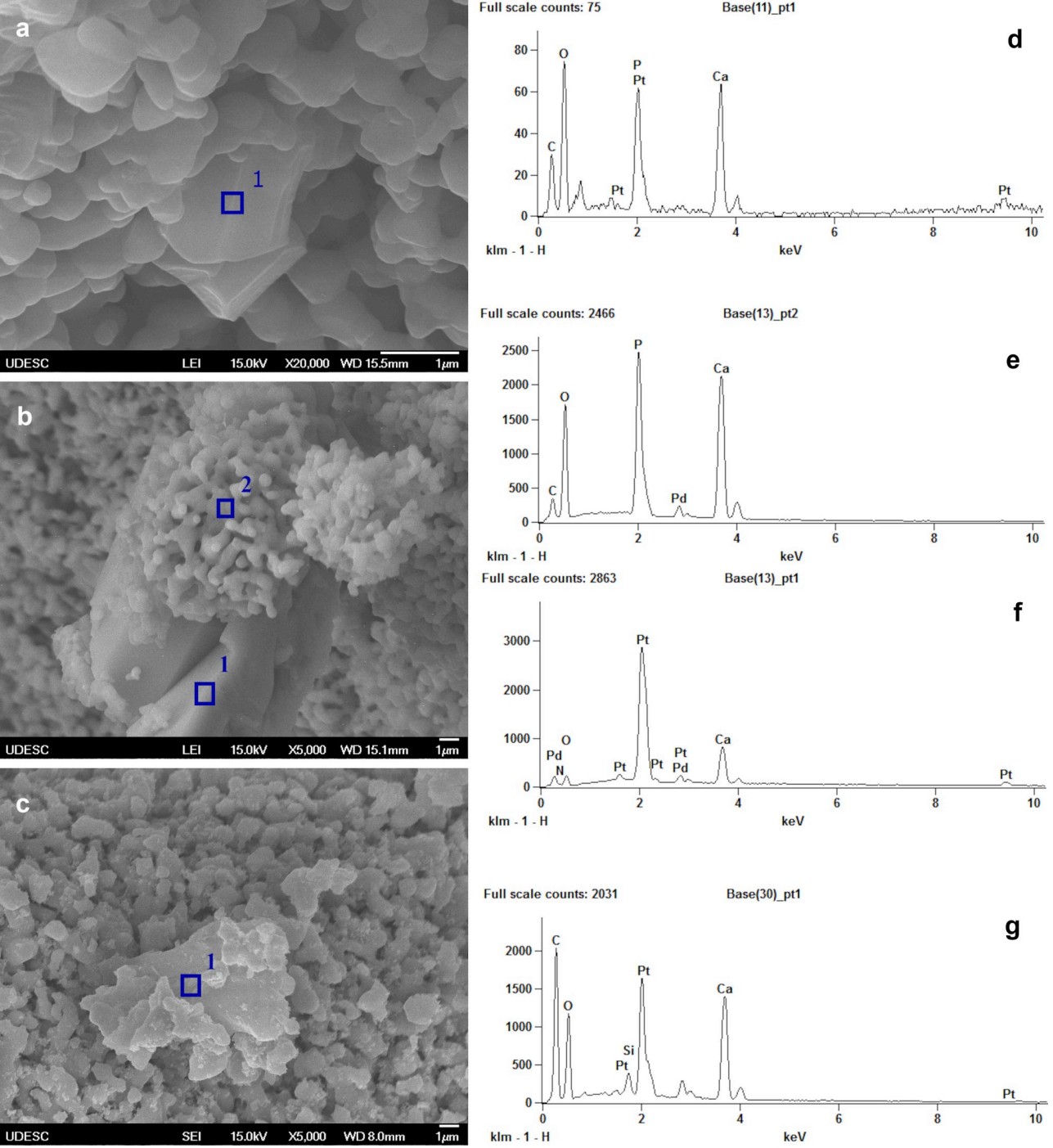

**Fig 3. Morphology and composition of carboplatin precipitates on the calcium phosphate surface.** Scanning electron microscopy analysis of the biomaterials-carboplatin loading: (a) well-defined micrometric carboplatin precipitated in HA-carboplatin 60 mg/g surface; (b) carboplatin precipitate and superposed calcium phosphate biomaterial in BCP-carboplatin 50 mg/g; (c) not well-defined boundary of carboplatin in β-TCP/SiO$_2$ nanocomposite 70 mg/g; energy dispersive spectroscopy (EDS) analysis detected peaks of marked regions: (d) the EDS analysis of point 1 in HA-carboplatin; (e) the EDS analysis of point 2 and (f) of point 1 in BCP-carboplatin; (g) the EDS analysis of the β-TCP/SiO$_2$ nanocomposite.

correlation with the amount of carboplatin concentration, although the composition showed an increase in Pt amount with the increment of the dose loaded for most biomaterials.

**Table 1. The initial surface area of the calcium phosphate (CaP) biomaterials.**

| CaP | Surface area (m$^2$.g$^{-1}$) |
|---|---|
| HA | 2.47 |
| β-TCP | 1.02 |
| BCP | 1.95 |
| β-TCP/MgO | 1.45 |
| β-TCP/SiO$_2$ | 2.44 |

## Fourier transform infrared spectroscopy

The FT-IR spectroscopy is a powerful technique capable of characterizing CaP biomaterials, as well as platinum drugs and mannitol [50–53]. Fig 5 illustrates the FT-IR spectra of the drug-loaded CaP biomaterials compared with the carboplatin drug alone. The spectra of biomaterials presented characteristic absorption bands of HA or/and β-TCP. HA and BCP biomaterials exhibited absorption band (narrow peak) at 3570 and 632 cm$^{-1}$ assigned to apatitic OH$^-$ groups; the characteristic phosphate groups stretching were detected in ~1090–1030 cm$^{-1}$ (ν3) and 962 cm$^{-1}$ for HA-contain and 970 and 944 cm$^{-1}$ for β-TCP-contain biomaterials (ν1). The CaP biomaterials also presented absorption bands assigned to phosphate bending mode in ~600–550 cm$^{-1}$ (ν4) and stretching mode in 474 cm$^{-1}$ (ν2). The FT-IR spectra of CaP-carboplatin indicated the presence of carboplatin and mannitol as well. Mannitol was detected in a broad band from–OH group in ~3400 cm$^{-1}$ and a band from–CH stretching mode at 2940 cm$^{-1}$. The bands of carboplatin were the broad bands in ~3280 and 1640 cm$^{-1}$ assigned to stretching and bending mode of NH$_3$ group, respectively, as well as–CH$_2$ bending mode and–C-O stretching mode in 1380 and 1348 cm$^{-1}$. The high-intensity bands assigned from–C-O of mannitol first and second alcohol in 1087 and 1025 cm$^{-1}$, as well as the characteristic vibrational modes of platinum-ligands, were not detected in FT-IR spectra of CaP-carboplatin biomaterials.

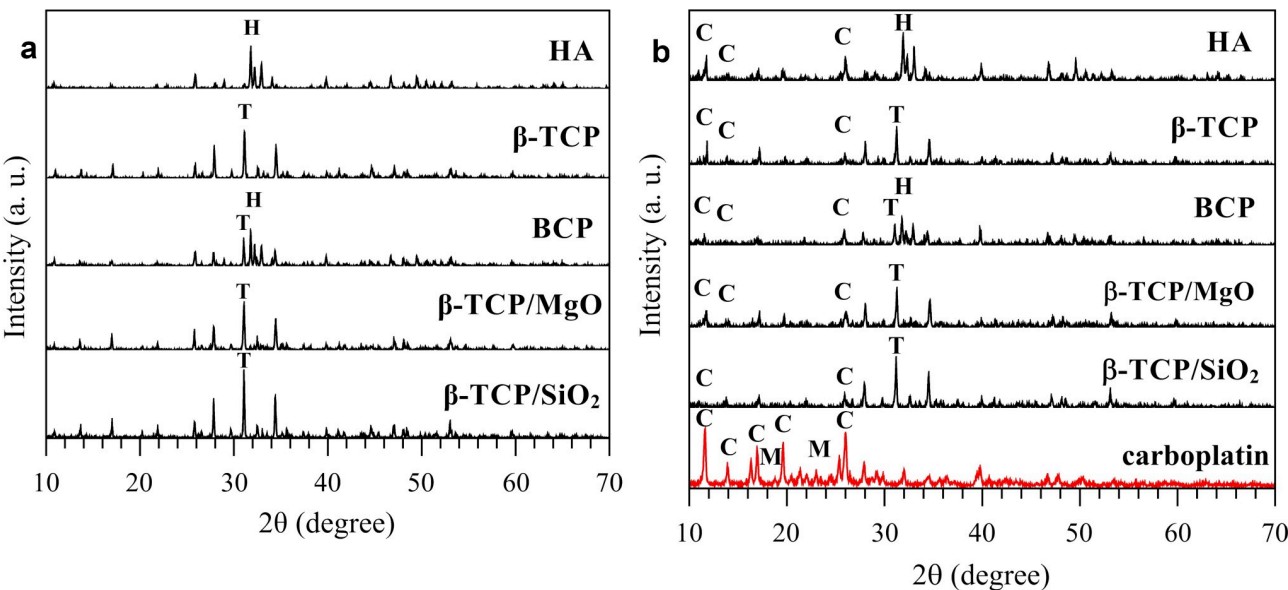

**Fig 4. X-ray diffraction analysis of the biomaterials before and after drug loading.** (a) detected characteristics hydroxyapatite (H) and β-TCP (T) peaks in the calcium phosphates (CaPs) before drug loading; (b) carboplatin (C) peaks in biomaterials after drug loading process in the 70 mg/g concentration, compared with the drug that presented carboplatin (C) and mannitol peaks (M).

**Table 2.  X-ray fluorescence spectroscopy of calcium phosphate (CaP)-carboplatin (wt%).**

| Biomaterial | Carboplatin (mg/g) | Ca | P | Si | Pt | Zr | Fe |
|---|---|---|---|---|---|---|---|
| **HA** | **50** | 80.220 | 12.096 | - | 7.626 | - | 0.058 |
|  | **60** | 84.671 | 5.416 | - | 9.823 | 0.023 | 0.068 |
|  | **70** | 85.163 | 3.775 | - | 10.995 | - | 0.067 |
| **β-TCP** | **50** | 82.924 | 10.898 | - | 6.083 | 0.039 | 0.056 |
|  | **60** | 80.464 | 12.918 | - | 6.545 | 0.020 | 0.053 |
|  | **70** | 84.088 | 8.118 | - | 7.730 | - | 0.058 |
| **BCP** | **50** | 82.924 | 10.898 | - | 6.083 | 0.039 | 0.056 |
|  | **60** | 80.874 | 10.951 | - | 8.113 | - | 0.062 |
|  | **70** | 81.737 | 12.290 | - | 5.914 | - | 0.060 |
| **β-TCP/MgO** | **50** | 83.737 | 7.631 | - | 8.549 | - | 0.082 |
|  | **60** | 82.796 | 5.825 | - | 11.284 | 0.025 | 0.070 |
|  | **70** | 84.364 | 4.946 | - | 10.617 | - | 0.073 |
| **β-TCP/SiO$_2$** | **50** | 79.536 | 10.796 | 3.568 | 6.045 | - | 0.055 |
|  | **60** | 77.244 | 11.095 | 3.265 | 8.298 | 0.038 | 0.059 |
|  | **70** | 78.682 | 13.339 | 3.563 | 4.361 | - | 0.055 |

### Raman spectroscopy

Raman spectroscopy is extensively used in the characterization of apatite compounds, which present Raman-active modes for $OH^-$ and $PO_4^{3-}$ groups [54,55]. The technique is also useful to detect platinum-ligands in the low-frequency range with high-intensity bands [51,56]. Fig 6 illustrates the Raman spectra of the 70 mg/g CaP-carboplatin biomaterials compared with carboplatin drug in the range of 3700–66 cm$^{-1}$ (Fig 6A) and the range of Raman scattering of platinum-ligands vibrational modes (700–150 cm$^{-1}$) (Fig 6B). All the CaP-carboplatin biomaterials presented the very strong band assigned to the asymmetric stretching mode of $PO_4^{3-}$ group (ν1) at 975–950 cm$^{-1}$, as well as the weak bands from the symmetric stretching mode (ν3). HA-contains biomaterials presented a band assigned to–OH stretching mode at ~3570 cm$^{-1}$ from apatite hydroxyl group. All the biomaterials presented superposed bands assigned to–$CH_2$ stretching modes from cyclobutane ring of carboplatin and–CH stretching modes from mannitol. β-TCP/MgO nanocomposite presented a very weak band assigned to $NH_3$ stretching mode from carboplatin (Fig 6A). The low-frequency range of CaP-carboplatin in all compositions is shown in Fig 6B. The $PO_4^{3-}$ bending modes were detected at 650–580 (ν4) and 470–450 cm$^{-1}$ (ν2). All the CaP-carboplatin Raman spectra presented bands at ~545 and ~471 cm$^{-1}$ assigned to Pt-N and Pt-O stretching modes, respectively. The weak intensity band corresponding to N-Pt-N bending mode at ~190 cm$^{-1}$ was detected in 50 and 60 mg/g HA-carboplatin, 60 mg/g HA/β-TCP-carboplatin, and 70 mg/g β-TCP-carboplatin only (Fig 6B). There was not detected any correlation between the Raman scattering intensity and the doses of carboplatin loaded.

### Ultraviolet-visible spectroscopy

UV-Vis spectroscopy is a simple and common technique used to measure carboplatin in previous studies [57,58]. The UV-Vis spectra of CaP-carboplatin biomaterials were compared with a carboplatin drug solution, a mannitol solution, and the blank release study containing only the CaP biomaterials, in the range of 190–400 nm. Fig 7A illustrates the CaP-carboplatin UV-Vis spectra in 60 minutes-time compared with carboplatin drug, and mannitol alone, both in 50 μg.mL$^{-1}$ solution. The UV-Vis spectra of all CaP-carboplatin biomaterials presented

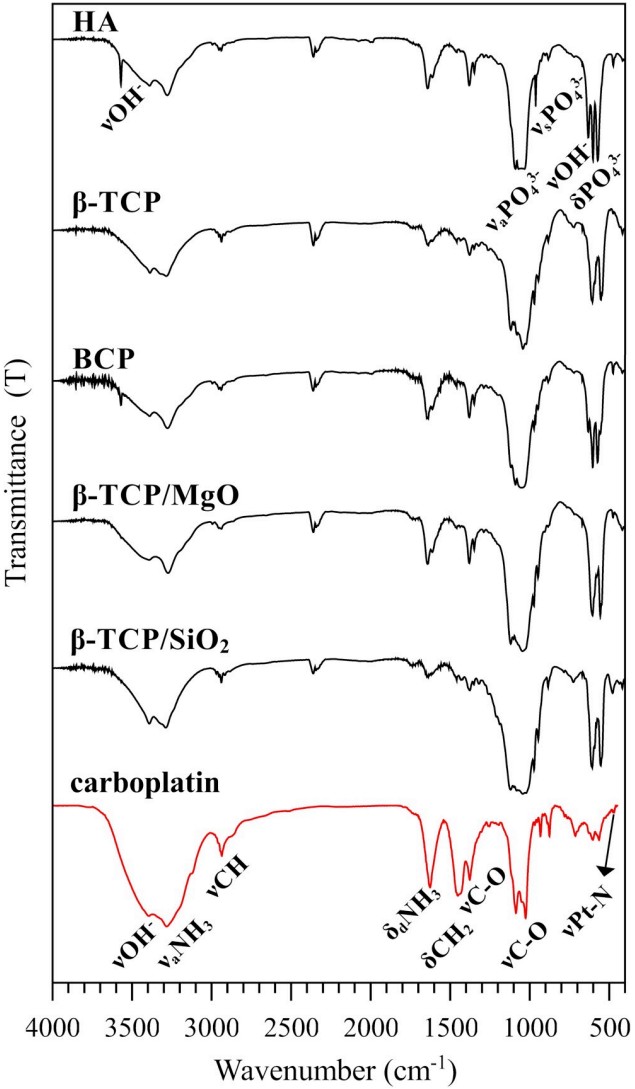

**Fig 5. Fourier transform infrared analysis of the biomaterials after the drug loading.** FTIR analysis of the 70 mg/g carboplatin-biomaterials compared with carboplatin drug alone (carboplatin and mannitol).

the same absorption behavior that carboplatin drug, meanwhile mannitol did not present any absorption and the substance has not considered in the release study. Fig 7B illustrates the UV-Vis absorption spectra of CaP-biomaterials of 60 minutes-release solution of blank release study. The highlighted line at the same wavelength of carboplatin measurement shows a slight absorption (absorbance scale is not the same of the one in Fig 7A), which was corrected from the carboplatin release study.

## Carboplatin loading on granular biomaterials

Table 3 shows the remaining weight of CaP-carboplatin after the high-vacuum process calculated by the CaP-carboplatin dried samples weight compared with the theoretical ones. The drug loading process using the vacuum method showed minimum losses, and the final CaP-carboplatin load biomaterials weights were 97.6 to 99.9% of the theoretical weight.

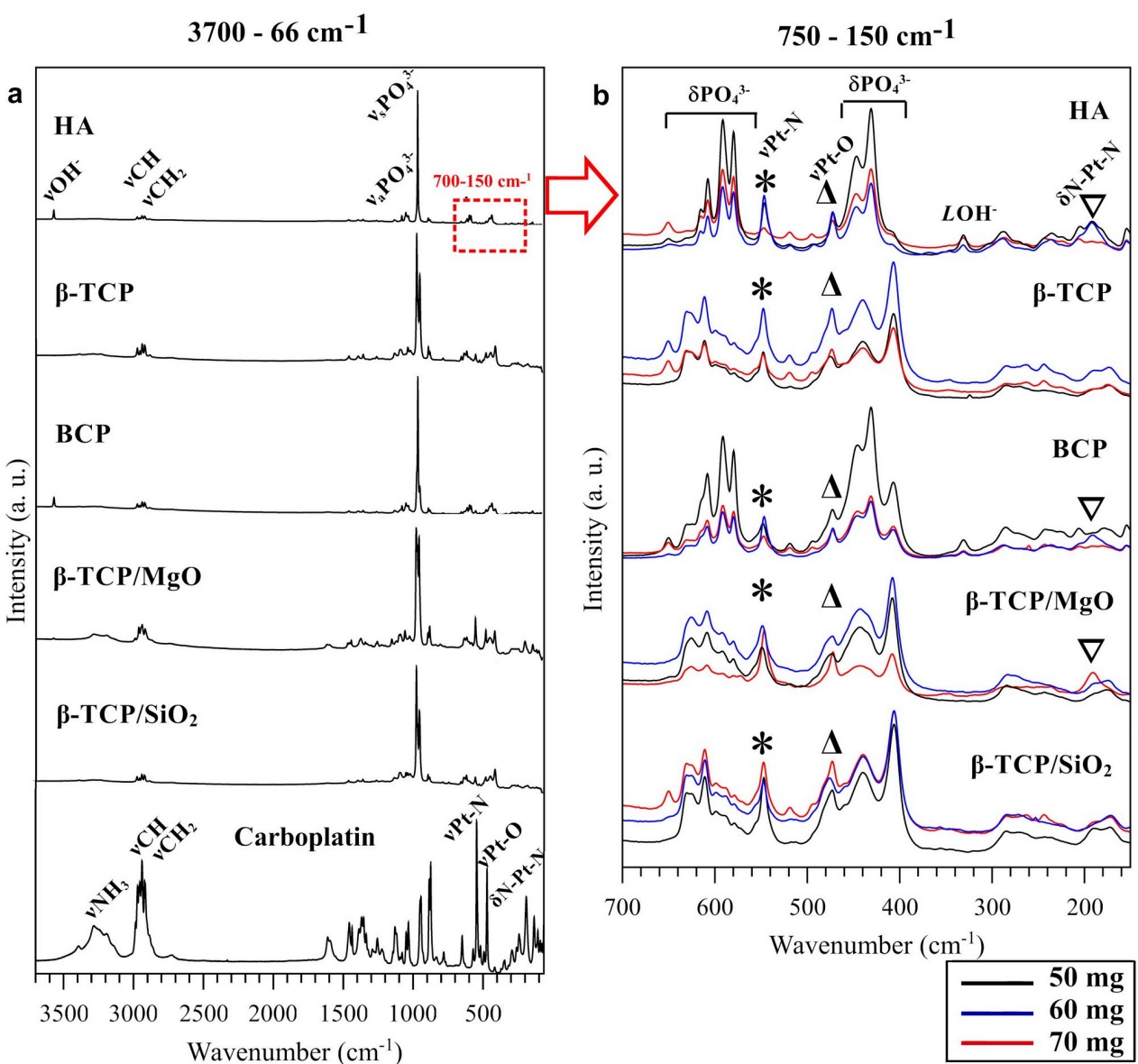

**Fig 6. Raman spectroscopy of the granulated calcium phosphate-carboplatin biomaterials and carboplatin drug alone (carboplatin and mannitol).** (a) High-intensity phosphate peaks of calcium phosphate biomaterials and carboplatin-drug detected peaks in the 70 mg/g concentration biomaterials; (b) 700–150 cm$^{-1}$ graphs with calcium phosphate peaks and characteristics platinum vibrational modes of carboplatin in 50 mg/g (black line), 60 mg/g (blue line) and 70 mg/g (red line) biomaterials.

### Carboplatin release

The *in-vitro* release curves of carboplatin from CaP granulated biomaterials in three different doses (50, 60 and 70 mg/g) are shown in Fig 8. The release profile exhibited a burst release of upon 60% of the drug-loaded in the first 15 minutes and a slow-release that maintained the drug rate during the study period. The CaP-carboplatin 50 mg/g (Fig 8A) release the concentration of the drug upon to 30 μg/mg of the biomaterials, for all compositions, that reached over 80% of the drug released during the study period. CaP-carboplatin 60 mg/g curves (Fig 8B) presented similar behavior with 80–90% of the drug released in 5 hours. The initial release for carboplatin 70 mg/g (Fig 8C) was upon 40 μg/mg for all compositions and over 70% was released

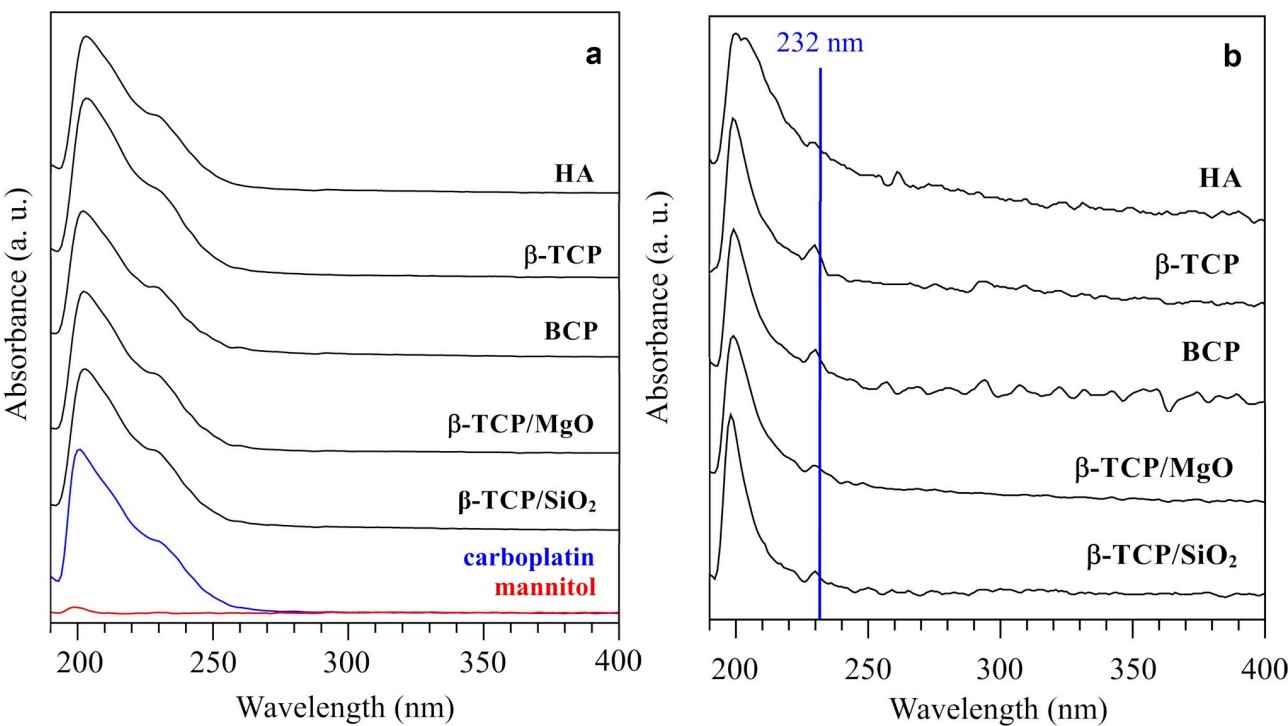

**Fig 7. Ultraviolet-visible (UV-Vis) spectroscopy analysis of release medium and blank tests.** (a) UV-Vis spectra of 60 minutes-release solution; 50 µg/mL carboplatin-drug (carboplatin and mannitol) solution (blue line), and 50 µg/mL mannitol solution alone (red line); (b) UV-Vis spectra of blank analysis with only biomaterials and blue line at 232 nm. Absorbance scales in Fig 7A and 7B are not the same.

after 5 hours. However, CaP composition did not show any influence on the rate of carboplatin release. The release behavior was more influenced by the doses previously loaded than the characteristics of the biomaterials. The amount of carboplatin released in the medium from 250 mg of the biomaterials was related to the previous doses loaded. The initial release was upon 8 mg (21.5 µmol) for 50 mg/g and 60 mg/g biomaterials-carboplatin and 10 mg (27 µmol) for 70 mg/g dose (S6 Fig). The pH measured after the release study was 7.4 ± 0.1 at 37°C.

**Table 3. Remaining weight of the CaP-carboplatin biomaterials after the load process by the high-vacuum method.**

| Biomaterial | Carboplatin (mg/g) | Theoretical weight (g) | CaP-carboplatin weight (g) | Remaining weight (%) |
|---|---|---|---|---|
| HA | 50 | 1.1000 | 1.0863 | 98.8 |
| | 60 | 1.1201 | 1.1118 | 99.3 |
| | 70 | 1.1401 | 1.1124 | 97.6 |
| β-TCP | 50 | 1.1000 | 1.0965 | 99.7 |
| | 60 | 1.1200 | 1.1018 | 98.4 |
| | 70 | 1.1400 | 1.1260 | 98.8 |
| BCP | 50 | 1.1000 | 1.0956 | 99.6 |
| | 60 | 1.1201 | 1.1064 | 99.8 |
| | 70 | 1.1400 | 1.1326 | 99.4 |
| β-TCP/MgO | 50 | 1.1001 | 1.0995 | 99.9 |
| | 60 | 1.1200 | 1.1100 | 99.1 |
| | 70 | 1.1400 | 1.1152 | 97.8 |
| β-TCP/SiO₂ | 50 | 1.1001 | 1.0853 | 98.6 |
| | 60 | 1.1201 | 1.1011 | 98.3 |
| | 70 | 1.1401 | 1.1191 | 98.2 |

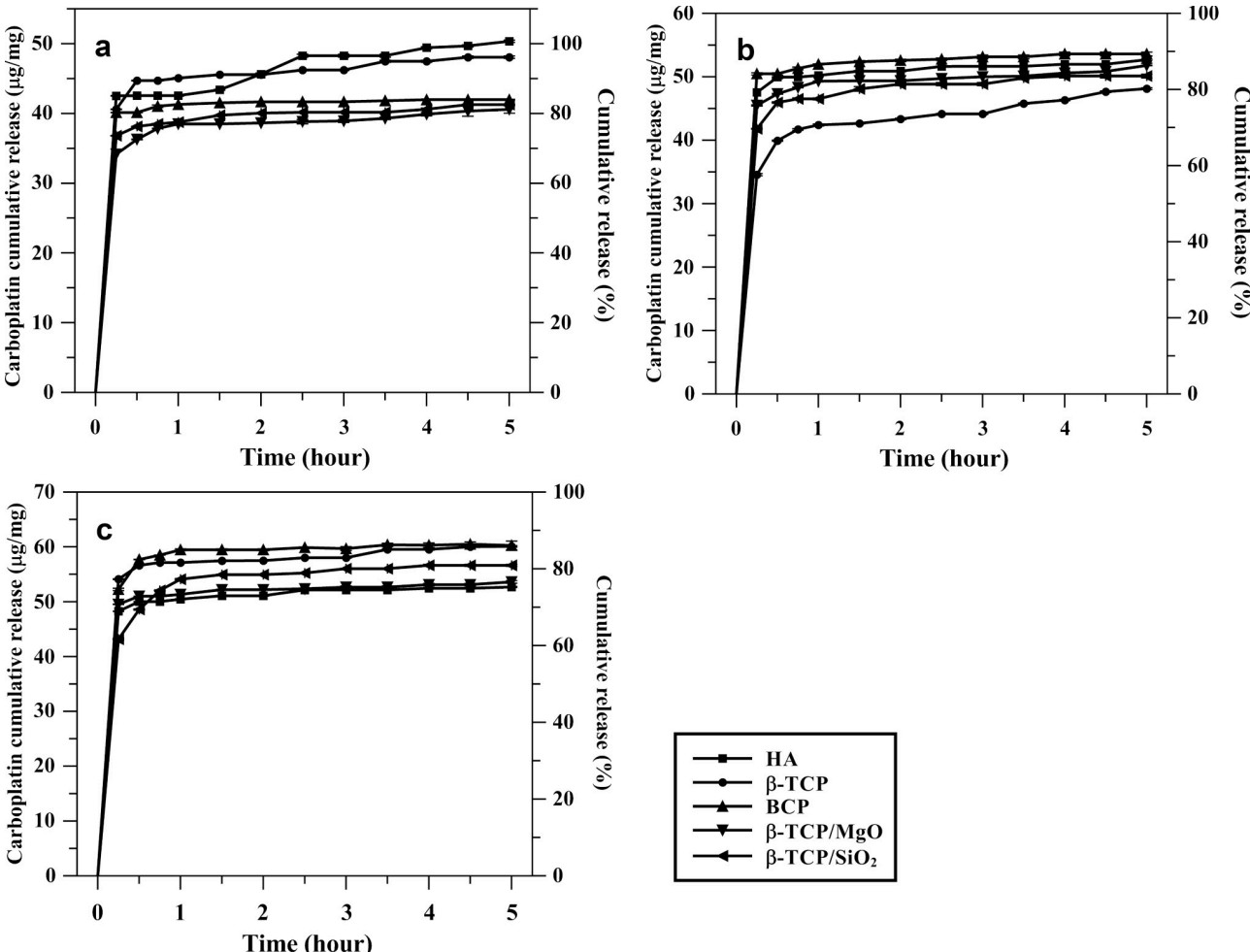

**Fig 8. Cumulative carboplatin release from biomaterials carboplatin-load in three different initial concentration.** (a) 50 mg/g; (b) 60 mg/g; (c) 70 mg/g.

## Discussion

Wet chemical precipitation of calcium phosphate powders in aqueous solution is widely used due to its simple experimentation and low cost [6]. Although water-soluble salts as a calcium source provide a homogeneous reaction medium, the presence of ions, such as acetate, chloride, or nitrate could be undesirable. The dissolution-precipitation method from calcium carbonate suspension as calcium source aims to reduce steps in the process such as the previous calcination of calcium carbonate to calcium oxide, washes, and purification [59]. The composition and morphological characteristics of calcium phosphate by aqueous precipitation using calcium carbonate, as the same as other calcium sources, are a result of the reaction conditions, especially temperature, acid addition rate, and reaction time [60–62]. This work used ultrasonic irradiation to achieve microporous powder of hydroxyapatite and β-TCP after calcination. The effect of ultrasonic irradiation has been associated with stoichiometric and crystalline calcium phosphate phase as demonstrated in another work [48]. The method induced disaggregation and deagglomeration of the particles by cavitation in the aqueous medium and improved the dissolution of calcium carbonate in the suspension.

The CaP granulated biomaterials of this work presented chemical composition and microstructure suitable to be used in bone regeneration as well as a matrix for drug delivery, due to their grain size, surface features, and the presence of the interconnected microporosity [2,4,45]. Calcium phosphates are widely known as biocompatible materials given their similar composition to biological apatites. However, different biological responses could be verified depending on the osteoinductive property and cell differentiation processes. Protein adsorption, biological apatite precipitation, and cell adhesion and differentiation have been related to the chemical composition, solubility, and surface characteristics such as microstructure and porosity [2,63,64]. CaPs also have been demonstrated biocompatibility as well as the osteoinductive mechanism by promoting inflammatory reaction and cell signaling *in vitro* and *in vivo* tests. Porosity, microstructure, and surface characteristics have been consistently related to cell adhesion in ceramic, metals, and polymers materials for tissue engineering [64–67]. HA, β-TCP and BCP ceramics have been extensively tested *in vitro* and *in vivo*. In the *in vivo* study of bone formation, Dalmonico et al. [2] showed that the high crystalline microporous granular CaPs biomaterials obtained by attrition milling presented phase composition and high open porosity, suitable to osseointegration and osteoinduction processes as well as biodegradability and bioactivity properties. Wang et al. [68] demonstrated inflammatory response and growth factors gene expression and osteoblastic differentiation in BCP ceramics as well as in their nanoparticles degradation products. The presence of other elements such as Si, Sr, and Mg have been associated with higher biological properties in CaPs biomaterials by influencing mineral metabolism and stimulating cellular activities [45,69]. MgO and $SiO_2$ have been added in β-TCP ceramics on purpose to modify the microstructure due to their ability to impair the grain growth in sintered materials [69,70]. In this paper, the high porous hydroxyapatite and β-TCP, which present higher solubility than HA, are used to produce five different compositions of microporous granular CaPs by the attrition milling process. The five different CaP compositions presented a specific surface area from 1.02 (β-TCP) to 2.47 $m^2.g^{-1}$ (HA). BCP showed an intermediate surface area due to the presence of both HA and β-TCP phases. In nanocomposites, the nanometric particles of the secondary phase increased the surface area to 1.45 and 2.44 $m^2.g^{-1}$ in β-TCP/MgO and β-TCP/$SiO_2$, respectively, and could be an interesting strategy to obtain biomaterials with a higher solubility than the one of HA without reducing the surface area, as in β-TCP materials only [63,71].

The SEM of the biomaterials after drug loading did not show any changes in the biomaterials microstructure and revealed micrometric precipitates of carboplatin, as well as mannitol on the CaPs surface. Backscattered electron analysis was used to highlight the platinum element in different matrices [72–75]. This technique uses the signal of the elastic interaction between the electron beam and the specimen. The BSE images present contrasts due to the proportional scattering cross-section with the mean atomic number (Z), in opposition to the secondary electron (SE) images that have low kinetic energy after escaping from the material surface [76]. The BSE image is dependent on the composition of the material and the condition analysis, such as the particle size, depth, energy, and incident angle of the primary beam [77]. In this work, precipitates from carboplatin and mannitol drug in the CaP surface were distinguished using BSE analysis, compared with SE images used in EDS analysis. The contrast between carboplatin and CaP matrices was slightly reduced compared with other works that highlighted the platinum element in carbon matrices [74,75]. The platinum content and the presence of other elements affect platinum identification in calcium phosphate-based matrices [73]. In this work the carboplatin composition Pt (Z = 78), O (Z = 8), N (Z = 7) and C (Z = 6) reduces the contrast between carboplatin precipitates in CaP matrices. Thus, BSE analysis was more effective in highlighting the micrometric precipitates of carboplatin. However, the contrast of drug precipitates of carboplatin and mannitol, which is composed of low Z atomic number, was remarkable.

SEM/EDS analysis detected Pt and N from carboplatin and demonstrates that drug precipitates were loaded in heterogeneous sizes and shapes in the biomaterials. The size and morphology of the drug in the CaPs surface are directly related to the incorporation method, which used predetermined drug concentration followed by solvent volatilization under vacuum.

The diffraction pattern of biomaterials before and after drug loading showed a crystalline pattern for carboplatin precipitates. High crystalline CaPs present lower loading capacity due to the better crystalline arrangement and less binding sites than non-crystalline materials, which present more surface defects and irregularities. Thus, high-crystalline CaPs are less reactive, but present more stability and less solubility [63,78,79]. The reduced peaks intensity of the CaPs after the drug loading process is attributed to the presence of carboplatin drug in the CaP surface [80].

Fluorescence X-ray analysis indicated the presence of platinum in all CaP biomaterials. Although XRF analysis has been described in the literature to quantity the among of drug adsorbed in mesoporous biomaterial, in this work the XRF results were not directly related to the loaded drug amount [1]. The XRF deviation found in some load concentrations was attributed to the heterogeneous precipitation in CaP surfaces. Spectroscopic techniques demonstrated the presence of the drug in CaP biomaterials. The presence of mannitol in the drug was detected by hydroxyl group vibrational mode in FT-IR and–CH stretching in Raman spectra [52,53,81]. Platinum vibrational modes at ~550–450 $cm^{-1}$ had better detection in Raman spectra than FT-IR technique due to the high intensity of the Raman peaks, while FT-IR spectra show high intensity of $\delta PO_4^{3-}$ from CaPs that covered the weak peaks of carboplatin in this region [50,51,56]. The presence of vibrational modes and peaks position of the drug and biomaterials indicated that the drug was attached to the CaP surface without any chemisorption process [15,80]. The minimum losses verified by load efficiency calculated (Eq 1) was expected for the loading methodology used and reflected only the losses in the process, not the adsorption capacity of calcium phosphates.

The proposed mechanism for carboplatin loading the granular biomaterials in this work is physisorption on the surface and microporosity. The weak Van der Waals interaction between carboplatin and CaPs was demonstrated by characterization methods applied and release behavior of carboplatin. The DRX, FTIR, and Raman results showed no crystallographic or chemical modifications of neither the matrix nor the drug after the loading process that suggests the reversible interaction. The drug-matrix chemical interaction was related to the appearance or disappearance of characteristic bands after drug loading as well as relevant changes in their position, as other studies reported [15,21,80].

The release of carboplatin from CaPs showed an initial fast release. In the first 15 minutes, more than 60% of carboplatin loaded was released into the medium by dissolving crystalline precipitates attached to the CaPs surface. The initial fast release followed by continuous delivery of carboplatin in the release study indicates an effect of the non-uniform distribution of the drug in the matrix. This non-uniform distribution of carboplatin onto the surface and microporosity was attributed to the evaporation at high-vacuum applied. The effect of the pressure can be assessed by comparing the ibuprofen loading on porous crystalline granular CaPs with similar surface area and porosity performed by Baradari et al. [21] and Chevalier et al. [44] using the impregnation method and evaporation under vacuum, respectively. The low pressure enhances the amount of the drug loaded inside the porosity and modified the release pattern [21,44]. The influence of the pressure on the amount of methotrexate loaded in porous HA blocks was also reported by Itokazu et al. [35]. However, the comparison between the results from different researches was limited not only by the distinct characteristics of the CaP matrices and drugs. The experimental conditions in the release experiments, such as release medium, pH, and volume medium, could influence the results since a standard test has not been established [4].

Additionally, the *in vitro* concentration verified in release tests cannot be extrapolated to an *in vivo* approach, especially for local delivery systems, due to the peculiarities of the biological or pathological environment. Nevertheless, release studies can provide an understanding of the release mechanism and drug/carrier interactions [4,21,24]. For example, the diffusion/dissolution model for non-uniform dispersed drug in non-swelling and non-degradable rigid polymer matrices was approached by Xiang and McHugh [82,83]. In calcium phosphate matrices, Shao et al. [25] also attributed the initial fast release followed by a sustained release in calcium phosphate biomaterials to a combination of different loading behavior from surface adsorption and the drug-loaded in the microporosity. The diffusion or dissolution from the surface of biomaterials attributed to the initial burst release of carboplatin, followed by a sustained release was verified even in polymer encapsulated with carboplatin drug [57,84]. Carboplatin and other platinum agents present a burst release profile in the CaPs biomaterials, as well as in other matrices, such as polymers, metallic materials, and mesoporous silica [58,85,86].

The phase composition and porosity of CaP biomaterials were related to *in vitro* and *in vivo* biodegradation results [2,45]. Although HA and β-TCP/SiO$_2$ nanocomposite presented higher surface areas among the CaPs biomaterials studied in this work, it was not verified a relationship between BET results and the release profile. The surface area has been related to the drug load capacity and release profile. However, the effect of the surface area in the amount of drug loaded has been considered more significant for large different surface values [43]. Considering surface normalized drug loading, other factors, such as drug-matrix interaction and environmental conditions of drug loading, had more influence on the drug load capacity [4,87,88]. The profile release in this study was attributed to the low interaction between CaPs and carboplatin, as well as the drug concentration and vacuum method used to drug loading [24]. The fast initial release of platinum agents demonstrated anticancer efficacy *in vitro* and *in vivo*. The effectiveness of local therapies using CaPs biomaterials depends on the concentration of the drug in the local site and the time of release. The loading process should consider the affinity between drugs and carriers. In some systems, especially drugs with poor affinity with matrix, the impregnation method could not provide drug amount to reach the concentration to therapeutic effect. Crystalline CaPs present more stability and good biological response but less surface area and available sites for drug interaction [79]. Therefore, alternative methods have been considered to improve drug amounts in CaPs matrices. Uchida et al. [36] demonstrated that an *in vivo* local delivery from CaPs blocks, filled with a predetermined amount of cisplatin, reached higher concentration in tumor site with a sustained release and reduced detection in plasma and other organs. Thus, local delivery systems based on porous CaPs could be an alternative to reduce side effects in traditional therapy [4,22,85,88].

The solvent evaporation under vacuum method allowed the incorporation of a predetermined amount of drug as in other works [25,44]. In the present study, the vacuum method proved to be useful for loading a low-interaction carboplatin drug to microporous CaP biomaterial.

## Conclusions

This study demonstrated that the evaporation at high-vacuum method allows the incorporation of a specific amount of drugs on the surface and microporosity of CaP biomaterials. On the contrary of impregnation, the vacuum method could be used to load low-interactions drugs. Well-crystalline CaPs present high stability and low solubility, being a suitable material for this purpose. After drug loading, the CaPs biomaterials showed heterogeneous carboplatin and mannitol precipitates from the drug composition used, attached to the CaPs surface. Carboplatin in CaPs was identified by the BSE, the spectroscopic and X-ray techniques. The crystalline precipitation of carboplatin in the CaPs surface was verified by XRD analysis. In this

work, carboplatin presented low interaction with CaPs biomaterials, verified by the absence of different vibrational modes and the release pattern in the *in-vitro* release study. The release pattern showed two stages. An initial burst release of the micrometric precipitates on the surface of CaPs was followed by a slow release of the carboplatin loaded in the microporosity. The different surface area, morphology, and composition of the CaPs used in this work did not influence neither the drug-loaded nor the release. Further *in vitro* studies are needed to evaluate the biocompatibility and biological properties of CaP biomaterials as well as the effect of carboplatin release from CaPs biomaterials in cancer cell lines.

## Supporting information

**S1 File. XRD dataset for biomaterials before drug loading.**
(XLSX)

**S2 File. XRD dataset for biomaterials after drug loading.**
(XLSX)

**S3 File. FTIR dataset for biomaterials after drug loading and carboplatin drug.**
(XLSX)

**S4 File. Raman dataset for biomaterials after drug loading and carboplatin drug.**
(XLSX)

**S5 File. UV-Vis dataset for blank study, release test, and drugs solutions.**
(XLSX)

**S1 Fig. UV-Vis calibration curve of carboplatin.**
(TIF)

**S2 Fig. Evolution of pH as a function of time during the synthesis of calcium phosphate with Ca/P molar ratio of 1.67 (left) and 1.5 (right).**
(TIF)

**S3 Fig. X-ray diffraction analysis of the hydroxyapatite and β-TCP powders used to produce the granular biomaterials.**
(TIF)

**S4 Fig. Fourier transform infrared analysis of the hydroxyapatite and β-TCP powders used to produce the granular biomaterials.**
(TIF)

**S5 Fig. Scanning electron microscopy analysis of the powders used to produce the granular biomaterials.** (a) hydroxyapatite and (b) β-TCP.
(TIF)

**S6 Fig. Concentration of carboplatin in release medium (100 mL) for 50, 60, and 70 mg/g biomaterials-carboplatin loading.** Amount of carboplatin released in μg and μmol for 250 mg of the biomaterials-carboplatin loading.
(TIF)

## Acknowledgments

The authors thank C-Labmu/PROPESP/UEPG for XRD, FT-IR, and Raman characterization in this work. The authors are also thankful for the Multi-User Facility infrastructure from Santa Catarina State University's Technological Sciences Center.

## Author Contributions

**Conceptualization:** Nelson Heriberto Almeida Camargo.

**Formal analysis:** Cristiane Savicki.

**Investigation:** Cristiane Savicki.

**Methodology:** Cristiane Savicki, Nelson Heriberto Almeida Camargo.

**Project administration:** Nelson Heriberto Almeida Camargo.

**Supervision:** Enori Gemelli.

**Writing – original draft:** Cristiane Savicki.

**Writing – review & editing:** Cristiane Savicki, Enori Gemelli.

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
