## [Decision Letter · Decision Letter 0]

18 Sep 2020

PONE-D-20-23966

Crystallization of carboplatin-loaded onto microporous calcium phosphate using high-vacuum method: Characterization and release study

PLOS ONE

Dear Dr. Savicki,

Thank you for submitting your manuscript to PLOS ONE. After careful consideration, we feel that it has merit but does not fully meet PLOS ONE’s publication criteria as it currently stands. Therefore, we invite you to submit a revised version of the manuscript that addresses the points raised during the review process.

We look forward to receiving your revised manuscript.

Kind regards,

Leming Sun, Ph.D.

Academic Editor

PLOS ONE

Journal Requirements:

Reviewers' comments:

Reviewer's Responses to Questions

**Comments to the Author**

1. Is the manuscript technically sound, and do the data support the conclusions?

Reviewer #1: Yes

Reviewer #2: Partly

2. Has the statistical analysis been performed appropriately and rigorously? 

Reviewer #1: Yes

Reviewer #2: N/A

3. Have the authors made all data underlying the findings in their manuscript fully available?

Reviewer #1: Yes

Reviewer #2: Yes

4. Is the manuscript presented in an intelligible fashion and written in standard English?

Reviewer #1: Yes

Reviewer #2: Yes

5. Review Comments to the Author

Reviewer #1: The authors report Crystallization of carboplatin-loaded onto microporous calcium phosphate using high-vacuum method: Characterization and release study. In this work, granulated CaPs were used as a matrix for loading the anticancer drug carboplatin using the high vacuum method. Morphological, chemical and surface modifications in the carboplatin-CaPs were investigated in detail. The characterization of the CaP-carboplatin biomaterials showed heterogeneous crystalline precipitation of the drug, and no morphological modifications of the CaPs biomaterials. The in vitro release profile of carboplatin from CaPs was evaluated by the ultraviolet-visible (UV-Vis) method. The curves showed a burst release of upon 60% of carboplatin loaded followed by a slow-release of the drug for the time of the study. The work is meaningful and can be considered to be published. However there still some key points should be further clarified, major revisions and re-valuation should be needed:

1. Some language errors should be carefully revised throughout the manuscript.

2. In the introduction part, it is better to list one or two examples to clarify the meaning and application of CaP delivery systems.

3. The authors use the vacuum pressures to improve the penetration of an anticancer drug in calcium phosphate blocks, however the further prospect is not clear, the authors should further clarify it.

4. In the material part, if the preparation of granular biomaterials adopts the previous preparation method, please provide the related literature.

5. As for the result part, the pH and solvent stability of calcium phosphate (CaP)-carboplatin should be determined.

6. The Biocompatibility of calcium phosphate (CaP)-carboplatin should be considered and discussed in detail.

7. As for Microstructure of the biomaterials, some literatures should be cited, such as J. Mater. Chem. A 2018 21216-21224; ACS Appl. Mat. Interfaces (2017) 32308-32315; J. Mater. Sci. (2019) 6719-6727; Mater. Lett. (2017) 82-84.

8. The comparison between the current work and previous examples should be considered in detail. The current work prospect is still not clear, which may be further explored through the comparison.

Reviewer #2: I only have few comments:

Why did the authors use the five different compositions? Are they tipical, or simply you wanted to test them? Please specify.

For Equation 1, Line 138, it should be "100" not "1", because the data were expressed as 100% not 1. Please check about that.

Although UV-Vis spectra method is simple but not accurate. It could be easily influenced by many factors. Hence, it is recommended to use other methods to verify.

Figure 8, drug release was done only in PBS but not physiological media. It is recommended to at least add proteins into the media to make the media more representative. Meanwhile, the authors only measured drug release once? There is no variation for all the data?

6. PLOS authors have the option to publish the peer review history of their article (what does this mean?). If published, this will include your full peer review and any attached files.

Reviewer #1: No

Reviewer #2: No

---

## [Author Response · Author response to Decision Letter 0]

20 Oct 2020

Dear. Leming Sun, Ph.D.

Academic Editor

PLOS ONE

The authors thank you very much for giving us a chance to revise our manuscript. The reviewers’ comments are valuable and very helpful for improving our research paper. We have carefully read all comments and have tried our best to revise the manuscript as per reviewers’ suggestions, which we hope to meet with acceptance requirements. We also thank the reviewers for the comments and suggestions.

Responses to Reviewer´s #1 Comments

1. Some language errors should be carefully revised throughout the manuscript.

The manuscript was revised. If the language is still not clear enough, as well as in the case of any specific typographical or grammatical error, we would be grateful if in the next round of revision the reviewer could indicate the sentences we should improve.

2. In the introduction part, it is better to list one or two examples to clarify the meaning and application of CaP delivery systems.

According to the reviewer´s suggestion, we added some examples in the manuscript to clarify the meaning and application of CaP delivery systems in Introduction (Manuscript, p. 3, lines 60, 62-67). It was pointed out that CaPs could be used as nanopowders or composites to produce delivery systems to be systemically administrated as alternatives to traditional drugs. However, granulated CaP, as well as cements and blocks, have been used as bone fillers. The addition of drugs in these materials is a way to deliver the drugs locally, avoiding the side effects of systemic administration. References were added/used in the manuscript to exemplify this question (BARADARI et al., 2012; BENEDETTI et al., 2015, 2016; CHINDAMO et al., 2020; DHATCHAYANI et al., 2020; KHALIFEHZADEH; ARAMI, 2020; LUCAS-APARICIO et al., 2020; MURATA et al., 2018; SWET et al., 2014). 

3. The authors use the vacuum pressures to improve the penetration of an anticancer drug in calcium phosphate blocks, however the further prospect is not clear, the authors should further clarify it.

The reviewer has correctly pointed out that the further prospect of the vacuum method can be clarified in the paper. According to the reviewer´s suggestion, we added some explanation in Introduction (p. 4, lines 92-95) to clarify that the impregnation method using high-vacuum pressure followed by solvent volatilization was applied in this research on purpose to improve the carboplatin loaded within the porous structure of granulated CaPs. It was further demonstrated in the results that the carboplatin-loaded in microporous structure affected the drug release due to the initial burst release of the drug precipitated on the surface followed by a sustained release from the dissolution of the drug from microporous (SHAO et al., 2012). 

References were added/used in the manuscript to exemplify this question (CHEVALIER et al., 2009; GBURECK et al., 2007). 

4. In the material part, if the preparation of granular biomaterials adopts the previous preparation method, please provide the related literature.

It was mentioned the related previous work in Materials and Methods – Synthesis and preparation of granular biomaterials (p. 5, lines 121-122), but we have added other literature (p. 6, line 133). We would like to point out that the Biomaterials Research Group at Santa Catarina State University used the method described in this paper to obtain granular biomaterials in different researches, with some modification. Hydroxyapatite and β-tricalcium phosphate powders were obtained by wet chemical methods using a calcium source and phosphoric acid (DALMÔNICO, 2015; DOROZHKIN, 2007; MUNARIN et al., 2015; RAYNAUD et al., 2002). The Biomaterials Group obtained CaP powders from calcium carbonate from calcareous marine sediment and H3PO4 raw materials by Silva et al. (2016). In the present paper, the research group described the modified method using the sonicated slurry of synthetic CaCO3 for the first time. Dalmonico et al. (2017) and Camargo et al. (2014) described the production of granular CaPs biomaterials.

References were added/used in the manuscript to exemplify this question (CAMARGO et al., 2014; DALMÔNICO et al., 2017; SILVA et al., 2016). 

5. As for the result part, the pH and solvent stability of calcium phosphate (CaP)-carboplatin should be determined.

As a part of the project, the in vitro solubility of CaP biomaterials was performed by soaking the materials in the phosphate buffer pH 7.4 at 37 ºC for four weeks and the materials presented stable weight stable (data not published). However, this study was carried out in Ф 10 mm pressed discs sintered at 1,100 ºC/2h but not in the granulated sintered CaPs. We have not included the data considering that the difference in surface area and microstructure of the materials not allowed the comparative analysis. 

The pH of CaP-carboplatin was measured in the released medium (phosphate buffer at 37 ºC) before and after the release test. The measured pH of 7.4 ± 0.1 results showed no significant deviation from the expected value. This information was added in Materials and Methods - Carboplatin release from the granular biomaterials (p. 7, lines 158-159) and Results - Carboplatin release section (p 16, line 377). Carboplatin is described as stable in aqueous solutions for 24 hours (https://pubchem.ncbi.nlm.nih.gov/compound/426756).

We would like to point out that the CaP-carboplatin stability was evaluated after the loading process by the X-ray diffraction and infrared spectroscopy analysis, as well as microscopy. In the release study, the UV-Vis spectroscopy was carried out and the same behavior was verified for CaP-carboplatin and carboplatin drug solution. The solubility and stability of calcium phosphate biomaterials were determined by some authors and crystalline CaPs are considered stable in pH and temperature conditions applied in this study (BERTAZZO et al., 2010; DOROZHKIN, 2007; WANG; NANCOLLAS, 2008). In vivo implants of sintered granular calcium phosphate biomaterials could present dissolution along the time. However, this process depends not only on the phase composition and surface characteristics but also on the biological environment and cell activity (DALMÔNICO et al., 2017; DOROZHKIN, 2007; DRAENERT; DRAENERT; DRAENERT, 2013). 

6. The Biocompatibility of calcium phosphate (CaP)-carboplatin should be considered and discussed in detail.

We agree that although the biocompatibility of granulated CaP-carboplatin was not the focus in this paper, this question should be considered due to the importance of this parameter to the applicability of the CaPs. Hence, the topic was discussed in Discussion (p. 17-18, lines 398-418). We also would like to point out that HA, β-TCP, and BCP granulated biomaterials, produced by the same method, have been tested in vivo (DALMÔNICO et al., 2017). The chemical composition, crystallinity, and microstructure of these biomaterials are quite similar to the CaPs used in this paper and they were compared with the previous study. The present study was designed to focus specifically on the synthesis and characterization of CaP-carboplatin biomaterials, but we have included your point as a consideration for future studies in Conclusions (page 23, line 540-541). References were added/used in the manuscript to exemplify this question (AGARWAL; GARCÍA, 2015; BERTAZZO et al., 2010; DALMÔNICO et al., 2017; DRAENERT; DRAENERT; DRAENERT, 2013; JIANG et al., 2017; SAMAVEDI; WHITTINGTON; GOLDSTEIN, 2013). 

7. As for Microstructure of the biomaterials, some literatures should be cited, such as J. Mater. Chem. A 2018 21216-21224; ACS Appl. Mat. Interfaces (2017) 32308-32315; J. Mater. Sci. (2019) 6719-6727; Mater. Lett. (2017) 82-84

The suggested literature was carefully considered. We compared the polymer materials in the suggested literature and cited ACS Appl. Mat. Interfaces (2017) 32308-32315 (p. 17, line 406) due to the high porosity and theoretical applicability as materials for tissue engineering. In other literature, however, the polymer materials are described to be suitable for another application but not as biomaterials. 

8. The comparison between the current work and previous examples should be considered in detail. The current work prospect is still not clear, which may be further explored through the comparison.

According to the reviewer´s suggestion, porous CaPs biomaterials for local delivery were compared to the biomaterials of the present study. A comparative analysis between characterization and the release profile described in the literature was added in Discussion (page 20, lines 469-473 and 478-484; page 21, lines 406-412; page 21, lines 505-505 and pages 21-22, lines 511-520). The results of the release study of the anti-inflammatory ibuprofen from porous crystalline granular β-TCP performed by Baradari et al. (2012) showed similar burst release of up to 70% of the drug in the first 15 minutes with a plateau of the release of about 70 min with almost 100% of the drug released in the first hour. The authors used the impregnation method, which depends on the drug/carrier interaction. Another study (CHEVALIER et al., 2009) loaded ibuprofen onto granulated β-TCP with similar composition, surface area, and porosity, using evaporation under vacuum. The comparative analyses showed that the vacuum method improves the contact between drug and microporosity and enhance the amount of the drug-loaded. The methodology applied also did not modify the fast initial release pattern but changed the percentage of the drug released at the same time. In our study, carboplatin presented similar reversible physisorption on CaPs matrices. Similarly, the release profile of CaP-carboplatin in this work also presented an initial fast release but showed a sustained release of the remaining carboplatin, suggesting a combination of the release of the drug from the surface followed by the drug in the microporosity as described by Shao et al (2012). References were added/used in the manuscript to exemplify this question (BARADARI et al., 2012; BARROUG et al., 2004; CHEVALIER et al., 2009; ITOKAZU et al., 1999; UCHIDA et al., 1992). 

Responses to Reviewer´s #2 Comments

Reviewer #2: I only have few comments:

1. Why did the authors use the five different compositions? Are they tipical, or simply you wanted to test them? Please specify.

Hydroxyapatite, β-TCP, and BCP are the common composition of calcium phosphate bioceramics. Hydroxyapatite presents well-known biocompatibility due to its chemical and crystallographic similarity with biological apatite. Hydroxyapatite is considered the less soluble non-substituted calcium phosphate. Therefore, the synthetic tricalcium phosphates have been used to increase de biological dissolution of bioceramics. HA presents fine microstructure and higher porosity, suitable to cell adhesion, than β-TCP. Biphasic HA/β-TCP ceramics have been studied on purpose to achieve better control over both solubility and microstructural characteristics (DALMÔNICO, 2015; DOROZHKIN, 2016; WANG; NANCOLLAS, 2008; ZHANG et al., 2013). 

The CaPs nanocomposites have been studied to modify chemical composition as well as the microstructure of the bioceramics. Silica and magnesium have been associated with increased biological response in bone regeneration processes (BOSE et al., 2011; PIETAK et al., 2007; PORTER et al., 2003; SILVA et al., 2016). In the present work, β-TCP/MgO and β-TCP /SiO2 nanocomposites were developed on purpose to achieve a higher porosity of the β-TCP phase. Carboplatin loading was tested in nanocomposites on purpose to verify the influence of the chemical composition in drug loading and release, due to the lack of information on the interaction between these specific compositions and carboplatin. The question was considered in the Introduction, and some words were added to the manuscript (page 5, lines 103-107) as well in the Discussion (Manuscript, pages 17-18, lines 412-418).

References were added/used in the manuscript to exemplify this question (BOSE et al., 2011; SILVA et al., 2016; WANG et al., 2017; ZHU et al., 2011). 

2. For Equation 1, Line 138, it should be "100" not "1", because the data were expressed as 100% not 1. Please check about that.

Equation 1 was originally based on weight loss equations described in the literature (TAN et al., 2009; TIAN; JIANG, 2018), where “1” was included to give results in terms of remaining weight instead of weight loss:

Equation 1:

Drug loading (%)=1- (W1-W2)/W1 x 100 (1)

In which:

W1 = theoretical weight calculated by the actual weight of the biomaterials and the drug content added, considering carboplatin and mannitol in its formulation; 

W2 = weight of the CaP-carboplatin samples after the drug loading.

For the sake of clarity, we modify Equation 1 as described as follow (GUAN et al., 2005):

Equation 1:

Remaining weight (%)=100 x W2/W1 (1)

In which:

W1 = theoretical weight calculated by the actual weight of the biomaterials and the drug content added, considering carboplatin and mannitol in its formulation; 

W2 = weight of the CaP-carboplatin samples after the drug loading.

We also modified some words in the original manuscript in the Materials and Methods (page 6, line 149) and the Results – Carboplatin loading on granular biomaterials (page 15, page 357). 

The title of Table 3 was modified as follows (page 15, lines 361-362). Table 3 data were not modified. 

Table 3. Remaining weight of the CaP-carboplatin biomaterials after the load process by the high-vacuum method.

3 Although UV-Vis spectra method is simple but not accurate. It could be easily influenced by many factors. Hence, it is recommended to use other methods to verify.

We agree with the reviewer that there are more accurate methods used in carboplatin detection, such as high-performance liquid chromatography (HPLC) (QU et al., 2017) or platinum detection using inductively coupled plasma techniques, such as ICP-AES, ICP-OES, and ICP-MS, (BAITUKHA et al., 2019; DOMÍNGUEZ-RÍOS et al., 2019; LELLI et al., 2016). However, UV-Vis techniques have been widely used to determine carboplatin in release studies from calcium phosphate biomaterials and other drug carriers (AKYUZ, 2020; BRAGTA et al., 2018; SHARMA; NASKAR; KUOTSU, 2020; SOUZA; ARDISSON; SOUSA, 2009; THAKUR et al., 2020). UV-Vis method has been also used as a technique in release studies of other drugs in porous drug carriers (BARADARI et al., 2012; BENEDINI et al., 2019; CHEVALIER et al., 2009; TSENG et al., 2015; ZHU et al., 2011). 

In our view, the UV-Vis method presents some advantages such as reduced sample manipulation and easy operation method. Some standard procedures such as baseline correction, linear calibration curve, as well as measurements of homogeneous samples in the calibration curve range, can minimize data errors. Our study attempted to address these issues by using a calibration curve with R = 0.99985 (S1_Sup. Information) and properly filtering the collected samples before measuring. We would like to point out that the UV-Vis method was used in this paper considering the influence of the CaPs biomaterials absorbance, which was measured and corrected before calculation, as described in Materials and Methods (page 7, lines 166-167) and Results – Ultraviolet-visible spectroscopy (page 14, lines 343-346) and showed in Figure 7 (page 15, line 350). It was also verified the influence of mannitol of the drug, that it was considered not relevant, as described in Results (page 15, lines 357-358). 

4. Figure 8, drug release was done only in PBS but not physiological media. It is recommended to at least add proteins into the media to make the media more representative. Meanwhile, the authors only measured drug release once? There is no variation for all the data?

In the consulted literature, some release studies were carried out in simulated body fluid or Dulbecco´s Modified Eagle Medium. Sodium chloride 0,9% solution or distilled water were also used (BAITUKHA et al., 2019; PARENT et al., 2017; PROKOPOWICZ, 2018; SOUZA; ARDISSON; SOUSA, 2009). However, phosphate buffers pH 7.4 at 37 ºC with no other addition has been used by most authors as a release medium for carboplatin, as well as other drugs (BARADARI et al., 2012; BENEDINI et al., 2019; BRAGTA et al., 2018; DALEY et al., 2018; DOMÍNGUEZ-RÍOS et al., 2019; QU et al., 2017; SHARMA; NASKAR; KUOTSU, 2020; THAKUR et al., 2020; TSENG et al., 2015). Parent et al. (2017) pointed out that the release experiments described in the literature present many differences in the release medium composition, pH, agitation, and volume, as well as the apparatus used in the experiments. In the present study, the release experiments were carried out in the same conditions of pH, medium composition, and temperature, as described for other authors. 

As mentioned in Methods – Carboplatin release from the granular biomaterials (page 7, line 156-161), the release tests were performed in sink conditions, avoiding supersaturation, and concentration gradient in the system. The parameters of the weight of CaP-carboplatin in the release test, agitation, release medium volume, and withdrawn volume samples were calculated considering the usual guidelines for sink conditions and the solubility of carboplatin under buffers solutions as well as other unpredictable factors such as system composition influence (SIEPMANN; SIEPMANN, 2020).

In our view, the release studies provide information about the release mechanism and drug/carrier interaction but cannot be extrapolated to an in vivo approach, especially in the case of local delivery matrices. For this reason, the focus of this work was on the mechanism verified and the interaction between carboplatin and CaPs matrices. Thus, each CaP-carboplatin juncture was measured in one experiment. Each withdrawn sample was measured in triplicate to obtain the correct absorbance, which explains the small variation in the data. For the reasons previously mentioned, the in vitro release tests in this work were reported in terms of percentage, and the concentration achieved in the initial release was mentioned just to give a parameter, but the focus was on the release profile. A similar approach has been used in release tests described in the literature, due to the several points measured in experiments, especially in the case of long-term release profiles as well as researches that tested several conditions of materials composition, drugs, or environmental release medium. In the literature, the following research papers could be cited: (ALINAVAZ et al., 2021; BAITUKHA et al., 2019; DADASHI; BODDOHI; SOLEIMANI, 2019; DOADRIO et al., 2015; IONITA et al., 2017; LEGNOVERDE; BASALDELLA, 2016; LIU et al., 2005; MEDERLE et al., 2016; ZHAI; LI, 2019).

Some explanations about the meaning and limitations of the in vitro release tests as well as challenge of data extrapolation to in vivo behavior were added to the original manuscript in the Discussion (pages 20-21, lines 484-491). 

References were added/used in the manuscript to exemplify this question (BARADARI et al., 2012; MARQUES et al., 2016; PARENT et al., 2017).

References

AGARWAL, R.; GARCÍA, A. J. Biomaterial strategies for engineering implants for enhanced osseointegration and bone repair. Advanced Drug Delivery Reviews, v. 94, p. 53–62, 2015. 

AKYUZ, L. An imine based COF as a smart carrier for targeted drug delivery: From synthesis to computational studies. Microporous and Mesoporous Materials, v. 294, p. 109850, 2020. 

ALINAVAZ, S. et al. Hydroxyapatite (HA)-based hybrid bionanocomposite hydrogels: ciprofloxacin delivery, release kinetics and antibacterial activity. Journal of Molecular Structure, v. 1225, p. 129095, 2021. 

BAITUKHA, A. et al. Optimization of a low pressure plasma process for fabrication of a drug delivery system (DDS) for cancer treatment. Materials Science & Engineering C, v. 105, p. 110089, 2019. 

BARADARI, H. et al. Calcium phosphate porous pellets as drug delivery systems: Effect of drug carrier composition on drug loading and in vitro release. Journal of the European Ceramic Society, v. 32, n. 11, p. 2679–2690, 2012. 

BARROUG, A. et al. Interactions of cisplatin with calcium phosphate nanoparticles: In vitro controlled adsorption and release. Journal of Orthopaedic Research, v. 22, p. 703–708, 2004. 

BENEDETTI, M. et al. Metalated nucleotide chemisorption on hydroxyapatite. Journal of Inorganic Biochemistry, v. 153, p. 279–283, 2015. 

BENEDETTI, M. et al. Adsorption of the cis-[Pt(NH3)2(P2O7)]2-(phosphaplatin) on hydroxyapatite nanocrystals as a smart way to selectively release activated cis-[Pt(NH3)2Cl2] (cisplatin) in tumor tissues. Journal of Inorganic Biochemistry, v. 157, p. 73–79, 2016. 

BENEDINI, L. et al. Adsorption/desorption study of antibiotic and anti-inflammatory dugs onto bioactive hydroxyapatite nano-rods. Materials Science & Engineering C, v. 99, p. 180–190, 2019. 

BERTAZZO, S. et al. Hydroxyapatite surface solubility and effect on cell adhesion. Colloids and Surfaces B: Biointerfaces, v. 78, p. 177–184, 2010. 

BOSE, S. et al. Understandin in vivo response and mechanical property variation in MgO, SrO and SiO2 doped β-TCP. Bone, v. 48, p. 1282–1290, 2011. 

BRAGTA, P. et al. Intratumoral administration of carboplatin bearing poly (ε-caprolactone) nanoparticles amalgamated with in situ gel tendered augmented drug delivery, citotoxicity, and apoptosis in melanoma tumor. Colloids and Surfaces B: Biointerfaces, v. 166, p. 339–348, 2018. 

CAMARGO, N. H. A. et al. Characterization of three calcium phosphate microporous granulated bioceramics. Advanced Materials Research, v. 936, p. 687–694, 2014. 

CHEVALIER, E. et al. Comparison of low-shear and high-shear granulation processes: Effect on implantable calcium phosphate granule properties. Drug Delelopment and Industrial Pharmacy, v. 35, n. 10, p. 1255–1263, 2009. 

CHINDAMO, G. et al. Bone diseases: current approach and future perspectives in drug delivery systems for bone target therapeutics. Nanomaterials, v. 10, n. 875, p. 1–35, 2020. 

DADASHI, S.; BODDOHI, S.; SOLEIMANI, N. Preparation, characterization, and antibacterial effect of doxycycline loaded kefiran nanofibers. Journal of Drug Delivery Science and Technology, v. 52, p. 979–985, 2019. 

DALEY, E. et al. Characterization of doxycycline-loaded calcium phosphate cement for treatment of aneurysmal bone cysts. Journal of Materials Science: Materials in Medicine, v. 29, n. 109, p. 1–6, 2018. 

DALMÔNICO, G. M. L. Elaboração e caracterização de biomateriais granulados microporosos de fosfatos de cálcio: Teste in vivo em ovinos. [s.l.] 2015. 211 p. Tese (Doutorado em Ciência e Eng. Materiais) - Universidade do Estado de Santa Catarina, 2015.

DALMÔNICO, G. M. L. et al. An in vivo study on bone formation behavior of microporous granular calcium phosphate. Biomaterials Science, v. 5, n. 7, p. 1315–1325, 2017. 

DHATCHAYANI, S. et al. Effect of curcumin sorbed selenite substituted hydroxyapatite on osteosarcoma cells: An in vitro study. Journal of Drug Delivery Science and Technology, v. 60, n. May, p. 101963, 2020. 

DOADRIO, A. L. et al. Use of anodized titanium alloy as drug carrier: Ibuprofen as model of drug releasing. International Journal of Pharmaceutics, v. 492, p. 207–212, 2015. 

DOMÍNGUEZ-RÍOS, R. et al. Cisplatin-loaded PLGA nanoparticles for HER2 targeted ovarian cancer therapy. Colloids and Surfaces B: Biointerfaces, v. 178, p. 199–207, 2019. 

DOROZHKIN, S. V. Calcium orthophosphates. J Mater Sci, v. 42, p. 1031–1095, 2007. 

DOROZHKIN, S. V. Biphasic, triphasic, and multiphasic calcium orthophosphates. Advanced Ceramics, v. 8, p. 33–95, 2016. 

DRAENERT, M.; DRAENERT, A.; DRAENERT, K. Osseointegration of hydroxyapatite and remodeling-resorption of tricalciumphosphate ceramics. Microscopy Research and Technique, v. 76, n. 4, p. 370–380, 2013. 

GBURECK, U. et al. Low temperature direct 3D printed bioceramics and biocomposites as drug release matrices. Journal of Controlled Release, v. 122, p. 173–180, 2007. 

GUAN, J. et al. Preparation and characterization of highly porous, biodegradable polyurethane scaffolds for soft tissue applications. Biomaterials, v. 26, n. 18, p. 3961–3971, 2005. 

IONITA, D. et al. Activity of vancomycin release from bioinspired coatings of hydroxyapatite or TiO2 nanotubes. International Journal of Pharmaceutics, v. 517, p. 296–302, 2017. 

ITOKAZU, M. et al. Local drug delivery system using ceramics: Vacuum method for impregnating a chemotherapeutic agent into a porous hydroxyapatite block. Journal of Materials Science: Materials in Medicine, v. 10, n. 4, p. 249–252, 1999. 

JIANG, S. et al. Ultralight, thermally insulatin, compressible polyimide fiber assembled sponges. ACS Applied Materials and Interfaces, v. 9, p. 32308–32315, 2017. 

KHALIFEHZADEH, R.; ARAMI, H. Biodegradable calcium phosphate nanoparticles for cancer therapy. Advances in Colloid and Interface Science, v. 279, p. 102157, 2020. 

LEGNOVERDE, M. S.; BASALDELLA, E. I. Influence of particle size on the adsorption and release of cephalexin encapsulated in mesoporous silica SBA-15. Materials Letters, v. 181, p. 331–334, 2016. 

LELLI, M. et al. Hydroxyapatite nanocrystals as a smart, pH sensitive, delivery system for kiteplatin. Dalton Transactions, v. 45, n. 33, p. 13187–13195, 2016. 

LIU, T. Y. et al. On the study of BSA-loaded calcium-deficient hydroxyapatite nano-carriers for controlled drug delivery. Journal of Controlled Release, v. 107, n. 1, p. 112–121, 2005. 

LUCAS-APARICIO, J. et al. Silicon-calcium phosphate ceramics and silicon-calcium phosphate cements: Substrates to customize the release of antibiotics according to the idiosyncrasies of the patient. Materials Science and Engineering C, v. 106, p. 110173, 2020. 

MARQUES, C. F. et al. Insights on the properties of levofloxacin-adsorbed Sr- and Mg-doped calcium phosphate powders. J Mater Sci: Mater Med, v. 27, n. 123, p. 2–12, 2016. 

MEDERLE, N. et al. Innovative biomaterials based on collagen-hydroxyapatite and doxycycline for bone regeneration. Advances in Materials Science and Engineering, v. 2016, n. ID 3452171, p. 1–5, 2016. 

MUNARIN, F. et al. Micro- and nano-hydroxyapatite as active reinforcement for soft biocomposites. International Journal of Biological Macromolecules, v. 72, p. 199–209, 2015. 

MURATA, T. et al. Evaluation of a new hydroxyapatite nanoparticle as a drug delivery system to oral squamous cell carcinoma cells. Anticancer Research, v. 38, n. 12, p. 6715–6720, 2018. 

PARENT, M. et al. Design of calcium phosphate ceramics for drug delivery applications in bone diseases: A review of the parameters affecting the loading and release of the therapeutic substance. Journal of Controlled Release, v. 252, p. 1–17, 2017. 

PIETAK, A. M. et al. Silicon substitution in the calcium phosphate bioceramics. Biomaterials, v. 28, p. 4023–4032, 2007. 

PORTER, A. E. et al. Comparison of in vivo dissolution processes in hydroxyapatite and silicon-substituted hydroxyapatite bioceramics. Biomaterials, v. 24, p. 4609–4620, 2003. 

PROKOPOWICZ, M. Characterization of low-dose doxorubicin-loaded silica-based nanocomposites. Applied Surface Science, v. 427, p. 55–63, 1 jan. 2018. 

QU, W. et al. EpCAM antibody-conjugated mesoporous silica nanoparticles to enhance the anticancer efficacy of carboplatin in retinoblastoma. Materials Science and Engineering C, v. 76, p. 646–651, 2017. 

RAYNAUD, S. et al. Calcium phosphate apatites with variable Ca/P atomic ratio I. Synthesis, characterisation and thermal stability of powders. Biomaterials, v. 23, n. 4, p. 1065–1072, 2002. 

SAMAVEDI, S.; WHITTINGTON, A. R.; GOLDSTEIN, A. S. Calcium phosphate ceramics in bone tissue engineering: A review of properties and their influence on cell behavior. Acta Biomaterialia, v. 9, p. 8037–8045, 2013. 

SHAO, F. et al. Ibuprofen loaded porous calcium phosphate nanospheres for skeletal drug delivery system. Journal of Materials Science, v. 47, n. 2, p. 1054–1058, 2012. 

SHARMA, S.; NASKAR, S.; KUOTSU, K. Metronomic chemotherapy of carboplatin-loaded PEGylated MWCNTs: synthesis, characterization and in vitro toxicity in human breast cancer. Carbon Letters, v. 30, n. 4, p. 435–447, 2020. 

SIEPMANN, J.; SIEPMANN, F. Sink conditions do not guarantee the absence of saturation effectsInternational Journal of Pharmaceutics, 2020. Disponível em: <https://doi.org/10.1016/j.ijpharm.2019.119009>

SILVA, D. F. et al. Characterization of mesoporous calcium phosphates from calcareous marine sediments containing Si, Sr and Zn for bone tissue engineering. Journal of Materials Chemistry B, v. 4, p. 6842, 2016. 

SOUZA, K. C.; ARDISSON, J. D.; SOUSA, E. M. B. Study of mesoporous silica/magnetite systems in drug controlled release. Journal of Materials Science: Materials in Medicine, v. 20, n. 2, p. 507–512, 2009. 

SWET, J. H. et al. A silica-calcium-phosphate nanocomposite drug delivery system for the treatment of hepatocellular carcinoma: In vivo study. J Biomed Mater Res - Part B, v. 102, n. 1, p. 190–202, 2014. 

TAN, R. et al. Preparation and characterization of an injectable composite. J Mater Sci: Mater Med, v. 20, n. 6, p. 1245–1253, 2009. 

THAKUR, S. et al. Thermosensitive hydrogel containing carboplatin loaded nanoparticles: A dual approach for sustained and localized delivery with improved safety and therapeutic efficacy. Journal of Drug Delivery Science and Technology, v. 58, p. 101817, 2020. 

TIAN, X.; JIANG, X. Preparing water-soluble 2, 3-dialdehyde cellulose as a bio-origin cross-linker of chitosan. Cellulose, v. 25, p. 987–998, 2018. 

TSENG, C. L. et al. Development of lattice-inserted 5-Fluorouracil-hydroxyapatite nanoparticles as a chemotherapeutic delivery system. Journal of Biomaterials Applications, v. 30, n. 4, p. 388–397, 2015. 

UCHIDA, A. et al. Slow release of anticancer drugs from porous calcium hydroxyapatite ceramic. Journal of Orthopaedic Research, v. 10, p. 440–445, 1992. 

WANG, J. et al. Role of biphasic calcium phosphate ceramic-mediated secretion of signalling molecules by macrophages in migration and osteoblastic differentiation of MSCs. Acta Biomaterialia, v. 54, p. 447–460, 2017. 

WANG, L.; NANCOLLAS, G. H. Calcium orthophosphates: Crystallization and dissolution. Chem. Rev., v. 108, p. 4628–4669, 2008. 

ZHAI, Q.-Z. Z.; LI, X.-D. D. Immobilization and sustained release of cefalexin on MCF nano-mesoporous material. Journal of Dispersion Science and Technology, v. 40, n. 11, p. 1675–1685, 2019. 

ZHANG, Y. et al. Dissolution properties of different compositions of biphasic calcium phosphate bimodal porous ceramics following immersion in simulated body fluid solution. Ceramics International, v. 39, p. 6751–6762, 2013. 

ZHU, M. et al. A mesoporous silica nanoparticulate/β-TCP/BG composite drug delivery system for osteoarticular tuberculosis therapy. Biomaterials, v. 32, p. 1986–1995, 2011.

---

## [Decision Letter · Decision Letter 1]

5 Nov 2020

Crystallization of carboplatin-loaded onto microporous calcium phosphate using high-vacuum method: Characterization and release study

PONE-D-20-23966R1

Dear Dr. Savicki,

We’re pleased to inform you that your manuscript has been judged scientifically suitable for publication and will be formally accepted for publication once it meets all outstanding technical requirements.

Kind regards,

Leming Sun, Ph.D.

Academic Editor

PLOS ONE

Additional Editor Comments (optional):

Thanks for your revision, According to the reviews' comments, I think this manuscript is ready for acceptance.

Reviewers' comments:

Reviewer's Responses to Questions

**Comments to the Author**

1. If the authors have adequately addressed your comments raised in a previous round of review and you feel that this manuscript is now acceptable for publication, you may indicate that here to bypass the “Comments to the Author” section, enter your conflict of interest statement in the “Confidential to Editor” section, and submit your "Accept" recommendation.

Reviewer #1: All comments have been addressed

Reviewer #2: All comments have been addressed

2. Is the manuscript technically sound, and do the data support the conclusions?

Reviewer #1: Yes

Reviewer #2: Yes

3. Has the statistical analysis been performed appropriately and rigorously? 

Reviewer #1: Yes

Reviewer #2: Yes

4. Have the authors made all data underlying the findings in their manuscript fully available?

Reviewer #1: Yes

Reviewer #2: Yes

5. Is the manuscript presented in an intelligible fashion and written in standard English?

Reviewer #1: Yes

Reviewer #2: Yes

6. Review Comments to the Author

Reviewer #1: (No Response)

Reviewer #2: The authors responded to all the comments well. The manuscript has been improved. The latest references are cited. Therefore should be accepted.

7. PLOS authors have the option to publish the peer review history of their article (what does this mean?). If published, this will include your full peer review and any attached files.

Reviewer #1: No

Reviewer #2: **Yes: **Yi Cao

---

## [Editor Report · Acceptance letter]

10 Nov 2020

PONE-D-20-23966R1 

Crystallization of carboplatin-loaded onto microporous calcium phosphate using high-vacuum method: Characterization and release study 

Dear Dr. Savicki:

I'm pleased to inform you that your manuscript has been deemed suitable for publication in PLOS ONE. Congratulations! Your manuscript is now with our production department. 

Kind regards, 

on behalf of

Dr. Leming Sun 

Academic Editor

PLOS ONE